# Towards Safe and Optimal Online Bidding: A Modular Look-ahead Lyapunov Framework

**Hengquan Guo**[1]***Haobo Zhang**[1]*, **Junwei Pan**[2]†, **Shudong Huang**[2], **Nianhua Xie**[2],
**Lei Xiao**[2], **Haijie Gu**[2], **Jie Jiang**[2], **Xin Liu**[1]†
[1] SIST, ShanghaiTech University, China, [2] Tencent Inc., China
{guohq,zhanghb2023,liuxin7}@shanghaitech.edu.cn, jonaspan@tencent.com

## Abstract

This paper studies online bidding subject to simultaneous budget and return-on-investment (ROI) constraints, which encodes the goal of balancing high volume and profitability. We formulate the problem as a general constrained online learning problem that can be applied to diverse bidding settings (e.g., first-price or second-price auctions) and feedback regimes (e.g., full or partial information), among others. We introduce L2FOB, a Look-ahead Lyapunov Framework for Online Bidding with strong empirical and theoretical performance. By combining optimistic reward and pessimistic cost estimation with the look-ahead virtual queue mechanism, L2FOB delivers safe and optimal bidding decisions. We provide adaptive guarantees: L2FOB achieves $O\big(\mathcal{E}_r(T,p) + (\nu^*/\rho)\mathcal{E}_c(T,p)\big)$ regret and $O\big(\mathcal{E}_r(T,p) + \mathcal{E}_c(T,p)\big)$ anytime ROI constraint violation, where $\mathcal{E}_r(T,p)$ and $\mathcal{E}_c(T,p)$ are cumulative estimation errors over $T$ rounds, $\rho$ is the average per-round budget, and $\nu^*$ is the offline optimal average reward. We instantiate L2FOB in several online bidding settings, demonstrating guarantees that match or improve upon the best-known results. These results are derived from the novel look-ahead design and Lyapunov stability analysis. Numerical experiments further validate our theoretical guarantees.

## 1 Introduction

Autobidding systems play an increasingly central role in the online advertising ecosystem, which channels hundreds of billions of dollars annually, continues to grow rapidly, and already accounts for the majority of total ad spend Aggarwal et al. (2024); eMarketer (2025). Operationally, a significant portion of digital ad inventory is allocated through real-time auctions that clear in milliseconds at an Internet scale. Platforms run billions of these auctions daily, making bidding strategy design a critical problem. Typically, an advertiser aims to maximize cumulative payoff over a fixed horizon (e.g., a day or a week) while respecting both a limited budget and a target return-on-investment (ROI) that captures profitability requirements during the period. With both budget and ROI constraints, online bidding must balance volume (e.g., impressions/clicks) and profitability: spending too fast sacrifices late-stage opportunities, spending too slowly leaves inventory unused, and pursuing volume at low ROI undermines profitability. This paper tackles this problem and presents a modular framework for safe and optimal online bidding under both budget and ROI constraints.

Most prior work studies online bidding under only budget constraints, where total expenditure is limited Balseiro & Gur (2019); Balseiro et al. (2020); Chen et al. (2024); Wang et al. (2023); Guo & Liu (2025). Budget constraints are comparatively easier to handle because costs are non-negative, so feasibility reduces to controlling a one-sided cumulative resource. However, modern advertisement markets require profitability control Golrezaei et al. (2021b), introducing ROI constraints with both packing and covering property: per-step violations can be offset over time, so long-horizon planning is essential. While Castiglioni et al. (2022) adopts an appropriate violation metric for ROI constraints, their analysis requires Slater's condition (existence of a strictly feasible policy). Such an assumption is difficult to certify for ROI constraints and incompatible with budget hard-stopping

---

*Equal contribution. † Corresponding author. Work done while Hengquan Guo was an intern at Tencent.

in many markets. In Castiglioni et al. (2025), an online bidding formulation incorporates both budget and ROI constraints, but their metric counts the number of constraint violations rather than the amount of violation. As a result, the proposed algorithm does not fully navigate the tradeoff between payoff and constraint violations. Moreover, existing approaches are tailored to specific environments and settings: for example, first-price or second-price auctions, bandits or convex optimization, and full information or partial information. There is no unified framework that captures the practical online bidding paradigm, which leads to the following:

*Can we design a unified, modular framework for safe and optimal online bidding under both budget and ROI constraints, with adaptive provable guarantees without Slater's condition?*

In this paper, we propose **L2FOB** (Look-ahead Lyapunov Framework for Online Bidding), which provides a positive answer to this question. Our main contributions are as follows:

- **Unified problem formulation.** We formulate online bidding as a general constrained online learning problem: an agent maximizes cumulative reward subject to budget and ROI constraints. Unlike prior work tied to specific auction models, we present a unified framework by treating reward $r(v, b)$ and cost $c(v, b)$ as general functions of context $v$ and bid $b$. As long as general online regression oracles for reward and cost are provided, as is common in the literature Foster & Rakhlin (2020); Slivkins et al. (2023), our framework is applied to different bidding environments with adaptive theoretical guarantees, providing a unified approach to online bidding.

- **Algorithm design.** L2FOB builds upon primal–dual methods for constrained online learning Yu & Neely (2020); Slivkins et al. (2023); Guo & Liu (2024), but adopts a Lyapunov perspective with look-ahead virtual queues and potential-shaped multipliers. To balance reward with budget and ROI constraints, L2FOB maintains two virtual queues, $Q_\rho$ (budget) and $Q_\gamma$ (ROI), which track real-time constraint satisfaction. Unlike these previous works where virtual queues directly accumulate violations, L2FOB only tracks "unsafe decisions" via the clip operator $(\cdot)^+$, yielding stricter safety and stronger guarantees. In the primal decision, rather than using fixed queue lengths, L2FOB introduces look-ahead queues that predict the induced violation before acting, enabling more precise violation control. We adopt a convex potential function to provide more flexible stability control. Together, these choices enable a modular framework with provably strong guarantees for online bidding optimization.

- **Theoretical guarantees.** L2FOB provides *general, strong, and adaptive* theoretical guarantees: it achieves $\mathcal{O}\big(\mathcal{E}_r(T, p) + (\nu^*/\rho)\mathcal{E}_c(T, p)\big)$ regret and $\mathcal{O}\big(\mathcal{E}_r(T, p) + \mathcal{E}_c(T, p)\big)$ *anytime* ROI violation, where $\mathcal{E}_r(T, p)$ and $\mathcal{E}_c(T, p)$ are the cumulative reward and cost estimation errors of the regression oracle, $\rho$ is the average budget per round, and $\nu^*$ is the offline optimal average reward. These results are established with hard stopping due to budget constraints and without assuming Slater's condition, providing guarantees that are more applicable in practice. We instantiate the framework for first-price auctions with budget constraints and for second-price auctions with both budget and ROI constraints, derive tighter guarantees, and apply it to constrained contextual bandits to match the best-known results. Detailed instantiations and comparisons are provided in Section 5. We run experiments in corresponding settings and demonstrate superior performance. Full details appear in Appendix D.

## 2 MODEL AND PRELIMINARIES

In this section, we formalize the general paradigm of online bidding under budget and return-on-investment (ROI) constraints and introduce the associated performance metrics.

**Safe Online Bidding under Constraints.** In this work, we study online auctions from the perspective of a single bidder interacting with a large population over $T$ rounds. In each round $t$, the bidder observes a context $v_t \in \mathcal{X}$ drawn from a known distribution, submits a bid $b_t \in \mathbb{R}^+$, and receives stochastic feedback $r_t = r(v_t, b_t) + \varepsilon_t^r$, $c_t = c(v_t, b_t) + \varepsilon_t^c$, where the unknown reward function and cost function are $r(v_t, b_t) \in [0, R_{\max}]$ and $c(v_t, b_t) \in [0, C_{\max}]$, $\varepsilon_t^{\{r,c\}}$ denote random noise terms.

To keep the formulation modular and widely applicable, we deliberately leave the forms of the context, reward, and cost unspecified. For example, the context could be the private value of the bidder and the reward is $r(v_t, b_t) = \mathbb{I}\{b_t \geqslant d_t\}(v_t - b_t)$, where $d_t$ is the highest competing bid,

under first-price payments the cost is $c(v_t, b_t) = \mathbb{I}\{b_t \geqslant d_t\}b_t$. Our framework also covers settings where $v_t$ encodes predictive features such as pCTR (predicted click-through rate) / pCVR (predicted conversion rate), in which case the reward and cost can be user-specified functions, thereby unifying a broad class of online bidding models within a single problem formulation. For each round $t \in [T]$, the objective is to choose bidding decisions that maximize cumulative expected revenue while ensuring the return-on-investment (ROI) remains at least $\gamma$ and the daily spend stays within a budget cap $B$. Formally, we consider the following constrained optimization problem:

$$\max_{\{b_t\}_{t=1}^T} \sum_{t=1}^T r(v_t, b_t) \quad \text{s.t.} \sum_{t=1}^T c(v_t, b_t) \leqslant B, \quad \sum_{t=1}^T r(v_t, b_t) \geqslant \gamma \sum_{t=1}^T c(v_t, b_t), \tag{1}$$

**Assumptions and Baselines.** We first introduce some basic notations and definitions. The optimal bid action b* is defined as the solution to the following offline problem:

$$\max_b \mathbb{E}\left[r(v, b)\right] \quad \text{s.t.} \; \mathbb{E}\left[c(v, b)\right] \leqslant \rho, \; \mathbb{E}\left[r(v, b)\right] \geqslant \gamma\mathbb{E}\left[c(v, b)\right]. \tag{2}$$

Let $\nu^* := \mathbb{E}\left[r(v_t, b^*)\right]$ denote the optimal expected reward, $\tau \leqslant T$ denote the stopping time at which the budget is exhausted under an algorithm, i.e.,

$$\tau = \operatorname*{argmin}_{\tau' \in [T]} \left\{\tau' \mid \sum_{t=1}^{\tau'} c(v_t, b_t) \geqslant B\right\}. \tag{3}$$

The regret of an algorithm over horizon $T$ is defined as

$$\text{Regret}(T) = T\nu^* - \mathbb{E}\left[\sum_{t=1}^{\tau} r(v_t, b_t)\right], \tag{4}$$

where $\{b_t\}_t$ denotes the bidding sequence of the decision maker.

While the bidding rule guarantees the budget constraint, we assess compliance with the ROI constraint via the following stronger anytime violation measures:

$$V_{\text{ROI}}(t) := \mathbb{E}\left[\gamma \sum_{s=1}^t c(v_s, b_s) - \sum_{s=1}^t r(v_s, b_s)\right]. \tag{5}$$

To establish a unified framework for the safe online bidding problem in diverse settings, we introduce the following general online regression oracle assumption.

**Assumption 1** *There exist online learning oracles* $\{\mathcal{O}\}_{r,c}$ *such that the reward and cost estimators* $\bar{r}_t(x, b)$ *and* $\bar{c}_t(x, b)$ *satisfy the following conditions with a high probability of at least* $1 - p$:

$$\omega = \left\{\begin{aligned} &|\bar{r}_t(v, b) - r(v, b)| \leqslant \varepsilon_t^r(p), \\ &|\bar{c}_t(v, b) - c(v, b)| \leqslant \varepsilon_t^c(p), \; v \in \mathcal{X}, \; b \in \mathbb{R}^+, \; t \in [T] \end{aligned}\right\},$$

*where we define the general cumulative error as* $\mathcal{E}_r(T, p) := \sum_{t=1}^T \varepsilon_t^r(p)$ *and* $\mathcal{E}_c(T, p) := \sum_{t=1}^T \varepsilon_t^c(p)$.

Our oracle assumption is mild and encompasses standard optimistic regression oracles. Classical optimism in the face of uncertainty designs already satisfy it Li et al. (2010); Foster et al. (2018); Foster & Rakhlin (2020); for constrained online learning/bidding, the estimators in Guo & Liu (2025); Wang et al. (2023); Vijayan et al. (2025); Castiglioni et al. (2025) also satisfy it. Richer function classes can also be accommodated by building confidence bounds via neural random features as in Zhou et al. (2020). Although the realized estimation error does depend on the data distribution induced by the policy, the oracle error bounds are uniform over all action selection rules satisfying the boundedness and noise conditions. Hence, these bounds are effectively independent of the specific sampling strategy.

Under Assumption 1, we can construct optimistic and pessimistic plug-in estimators as follows:

$$\hat{r}_t(v, b) := \bar{r}_t(v, b) + \varepsilon_t^r \quad \text{(optimistic / upper confidence bound reward)},$$
$$\check{c}_t(v, b) := \bar{c}_t(v, b) - \varepsilon_t^c \quad \text{(pessimistic / lower confidence bound cost)}.$$

On the high-probability event $\omega$ (which occurs with probability at least $1 - p$), the following results hold simultaneously for all $(v, b)$ and all $t \in [T]$:

$$0 \leqslant \hat{r}_t(v, b) - r(v, b) \leqslant 2\varepsilon_t^r, \; 0 \leqslant c(v, b) - \check{c}_t(v, b) \leqslant 2\varepsilon_t^c.$$

# 3 SAFE ONLINE BIDDING ALGORITHM

We propose L2FOB (Look-ahead Lyapunov Framework for Online Bidding), a *modular, safe, and optimal* online bidding framework under budget and ROI constraints. Our framework leverages online regression oracles to produce *optimistic* reward and *pessimistic* cost estimates. To enforce the constraints, L2FOB maintains two virtual queues, $Q_\rho$ and $Q_\gamma$, to track budget and ROI feasibility, and uses the convex potential function $\Phi(\cdot)$ to guarantee safety. At the core is a *look-ahead virtual-queue* mechanism that incorporates predicted one-step constraint violations to refine the reward-constraint tradeoff. Prior works require Slater's condition for ROI constraints, namely, the existence of a strictly feasible margin $\mathbb{E}[r(v,b)] \geqslant \gamma\mathbb{E}[c(v,b)] + \xi$, $\xi > 0$, which means the ROI target has slack. The designs of L2FOB allow us to provide provable safe and optimal guarantees without Slater's condition.

---

**Algorithm 1** Look-ahead Lyapunov Framework for Online Bidding (L2FOB)

---

**Require:** Convex potential $\Phi(\cdot) = (\cdot)^2$; initial estimators $\hat{r}_1(\cdot,\cdot)$ and $\check{c}_1(\cdot,\cdot)$, virtual queues $Q_\rho(0) = Q_\gamma(0) = 0$ and $\eta_\rho = \eta_\gamma = \sqrt{T}$.

1: **for** $t = 1$ to $T$ **do**
2:     **Observe:** bidding opportunity with context $v_t$ from the learning oracles.
3:     **Construct estimators:** $\hat{r}_t(\cdot,\cdot)$ and $\check{c}_t(\cdot,\cdot)$.
4:     **Look-ahead virtual queue updates:** for any $b \geqslant 0$,

$$Q_\rho(t,b) = Q_\rho(t-1) + \eta_\rho(\mathbb{E}_v[\check{c}_t(v,b)] - \rho)^+, \tag{6}$$

$$Q_\gamma(t,b) = Q_\gamma(t-1) + \eta_\gamma(\mathbb{E}_v[\gamma\check{c}_t(v,b) - \hat{r}_t(v,b)])^+. \tag{7}$$

5:     **Bid action:** choose $b_t$ as maximizer of

$$\hat{r}_t(v_t,b) - \Phi'(Q_\rho(t,b))\eta_\rho(\mathbb{E}_v[\check{c}_t(v,b)] - \rho)^+ - \Phi'(Q_\gamma(t,b))\eta_\gamma(\mathbb{E}_v[\gamma\check{c}_t(v,b) - \hat{r}_t(v,b)])^+. \tag{8}$$

6:     **Submit bid $b_t$:** observe reward $r_t$ and cost $c_t$.
7:     **Virtual queue updates:**

$$Q_\rho(t) = Q_\rho(t-1) + \eta_\rho(\mathbb{E}_v[\check{c}_t(v,b_t)] - \rho)^+, \tag{9}$$

$$Q_\gamma(t) = Q_\gamma(t-1) + \eta_\gamma(\mathbb{E}_v[\gamma\check{c}_t(v,b_t) - \hat{r}_t(v,b_t)])^+. \tag{10}$$

8:     **Estimators update:** update $\hat{r}_{t+1}$ and $\check{c}_{t+1}$.
9:     **Stopping rule:** if $\sum_{s=1}^t c(v_s, b_s) \geqslant B$ then **break**.
10: **end for**

---

- **Look-Ahead Virtual Queue Tracing.** The key design that distinguishes L2FOB from prior work is the look-ahead virtual queue tracing. While Slivkins et al. (2023); Han et al. (2023); Guo & Liu (2024; 2025) update dual variables only after observing cost feedback, L2FOB computes look-ahead virtual queues for each candidate bid before the decision. The queues act as dynamic pacing variables that modulate exploration and conservatism. By incorporating predicted violations into the queues, L2FOB internalizes feasibility at decision time rather than merely tracking the past, yielding more precise safety control. L2FOB passes the queues through a convex potential function $\Phi(\cdot)$ and uses its derivative as the time-varying multipliers $\lambda_\rho(t) := \Phi'(Q_\rho(t,b))$, $\lambda_\gamma(t) := \Phi'(Q_\gamma(t,b))$. The choice of Lyapunov function is often regarded as more of an art than a strict technique, with different designs yielding different analytical outcomes. For example, $\Phi(x)$ is set to be $e^x$ in Sinha & Vaze (2024) to achieve the best-known results in constrained online convex optimization. In this paper, we show that the classical choice $\Phi(x) = x^2$ is sufficient; however, we retain a tunable design to enable L2FOB to flexibly control constraint violations.

- **Safe Online Bidding Decision.** Given context $v_t$ and the optimistic and pessimistic estimators, L2FOB selects $b_t \in \mathbb{R}^+$ by greedily maximizing the surrogate objective in (8). This resembles a primal-dual step where the primal module maximizes an approximate Lagrangian

$$L(v_t,b) = r(v_t,b) - \lambda_\rho^*(t)\big(c(v_t,b) - \rho\big) - \lambda_\gamma^*(t)\big(\gamma c(v_t,b) - r(v_t,b)\big),$$

with $\lambda_\rho^*(t)$ and $\lambda_\gamma^*(t)$ denoting the offline optimal dual variables. However, L2FOB has the following key differences: First, we apply the clip $(\cdot)^+$ so that penalties activate only when the

estimated constraints are violated, which lets the algorithm focus on reward maximization when constraints appear satisfied and supports a more flexible penalty design. Second, L2FOB enforces constraints in a mean-field fashion, where $\mathbb{E}_v[\cdot]$ denotes expectation over the context marginal. This focuses the penalty on systematic risk rather than per-context noise and yields smoother safety control. Moreover, the dual approximators utilize a convex potential $\Phi(\cdot)$ in conjunction with a look-ahead virtual queue mechanism, which enhances violation detection and mitigates overly aggressive updates. Unlike Guo & Liu (2025); Wang et al. (2023); Castiglioni et al. (2025), which are tailored to specific problem environments and settings, L2FOB adapts broadly through the general online regression oracle assumption.

In summary, L2FOB is a modular framework that combines general optimistic and pessimistic estimation with look-ahead virtual queue tracing and potential-shaped multipliers. It can apply to various bidding environments by applying dedicated online regression oracles. The look-ahead design is indeed closely related in spirit to one-step model predictive control, where a Lyapunov function is used as a surrogate for long-term performance, and the controller chooses the current action by greedily optimizing a one-step prediction that combines immediate reward/cost with the change in this function Rawlings et al. (2020). The key difference is that the "state" of our Lyapunov function is given by virtual queues that track constraints rather than the plant state.

## 4    MAIN RESULTS AND PROOF SKETCH

In this section, we first provide the regret and violation guarantee of the L2FOB algorithm in the following main theorem.

**Theorem 1** *Under Assumption 1, the L2FOB framework achieves*

$$\text{Regret}(T) = O\left(\mathcal{E}_r(T,p) + \frac{\nu^*}{\rho}\mathcal{E}_c(T,p)\right), \ \text{V}_{\text{ROI}}(t) = O\left(\mathcal{E}_r(T,p) + \mathcal{E}_c(T,p)\right), \ \forall t \in [T].$$

To our knowledge, Theorem 1 proves the first *adaptive* guarantees on both regret and ROI violation for safe online bidding. The bounds scale with the oracle errors $\mathcal{E}_r(T,p)$ and $\mathcal{E}_c(T,p)$. Moreover, we provide a stronger anytime ROI violation that guarantees safety over the whole time horizon. Theorem 1 shows that any learning procedure satisfying Assumption 1 can be plugged into the framework. In Section 5, we instantiate Theorem 1 in the settings of Wang et al. (2023), Castiglioni et al. (2025), and Guo et al. (2025), demonstrating how the framework adapts to these environments. In the settings considered by Wang et al. (2023); Castiglioni et al. (2025), we show that L2FOB achieves improved theoretical guarantees via Theorem 1. In the constrained bandit setting of Guo et al. (2025), our results match the best-known guarantees and can be further tightened given sufficiently accurate learning oracles.

**Remark 1** *While we do not provide a strict zero-violation guarantee, all these results are obtained without assuming Slater's condition, which makes them more generally applicable. When Slater's condition does hold, a strict zero-violation guarantee can be enforced by solving a tightened problem with a safety margin (e.g., replacing $\gamma$ with $\gamma + \delta$ for some $\delta > 0$), as in Liu et al. (2021); Castiglioni et al. (2025). Even without such a tightening, our method already achieves low constraint violation by controlling the stricter metric $\mathbb{E}\left[\sum_{s=1}^{t}\left(\mathbb{E}_v[\gamma c(v,b_s) - r(v,b_s)]\right)^+\right]$, as established in the proofs in Section 4.1.*

### 4.1    PROOF OF THEOREM 1

In this section, we provide a proof sketch of Theorem 1, a detailed version can be found in Appendix E. We first present the following key lemma, which provides a unified bound on regret and Lyapunov drift and serves as the key for proving Theorem 1.

**Lemma 1** *Let $b_t$ be the bidding action of the L2FOB algorithm, then for any bidding action $b$, the following inequality holds*

$$\hat{r}_t(v_t, b) - \hat{r}_t(v_t, b_t) + \Phi(Q_\rho(t)) - \Phi(Q_\rho(t-1)) + \Phi(Q_\gamma(t)) - \Phi(Q_\gamma(t-1))$$
$$\leqslant 4\varepsilon_t^r + \Phi'(Q_\rho(t,b))\eta_\rho(\mathbb{E}_v[\check{c}_t(v,b)] - \rho)^+ + \Phi'(Q_\gamma(t,b))\eta_\gamma(\mathbb{E}_v[\gamma\check{c}_t(v,b) - \hat{r}_t(v,b)])^+,$$

We use a convex potential function $\Phi(\cdot)$ to provide more flexible control of the virtual queues. We adopt the most common quadratic function $\Phi(x) = x^2$ but allow it to be tunable to provide more flexible violation control. This design, together with the rectified constraint and the look-ahead virtual queues, yields the following lemma, which removes the need for Slater's condition (existence of a strictly feasible solution) assumed in Slivkins et al. (2023); Guo & Liu (2024; 2025). Note that Guo et al. (2025) circumvents Slater's condition through a mean-field approach that leverages the context distribution, but it focuses on contextual bandits with a single constraint. Through the above key lemma, we then derive the Lyapunov stability of L2FOB.

**Lyapunov Stability.** Lyapunov drift analysis has long been used to certify the stability of control policies in networking and operations research Hajek (1982); Eryilmaz & Srikant (2012), where stability is enforced by controlling queue lengths. In our setting, we leverage virtual queues to track the constraint violation over time. We adopt this perspective by modeling the budget and ROI constraints with the virtual queues $Q_\rho$ and $Q_\gamma$, which can be viewed as stochastic/Markovian processes. The following lemma, derived from Lemma 1, formalizes this stability property of virtual queues and provides an explicit bound on the virtual queues.

**Lemma 2** *Under the L2FOB algorithm, the following inequality holds for any $t \in [T]$,*
$$\mathbb{E}[Q_\rho(t)] = O(\sqrt{T}), \ \mathbb{E}[Q_\gamma(t)] = O(\sqrt{T}).$$

The lemma implies stability of the budget and ROI constraints under L2FOB over the whole horizon $T$. Based on it, we can now derive the corresponding regret and anytime ROI violation guarantees.

**Constraint Violation.** We begin with a simple observation. The relationship between queue lengths and cumulative violation follows directly from the update rules in (9) and (10).

$$\eta_\rho \sum_{t=1}^{\tau} (\mathbb{E}_v[\check{c}_t(v, b_t)] - \rho)^+ \leqslant Q_\rho(\tau), \ \eta_\gamma \sum_{s=1}^{t} (\mathbb{E}_v[\gamma \check{c}_s(v, b_s) - \hat{r}_s(v, b_s)])^+ \leqslant Q_\gamma(t), \qquad (11)$$

To derive a guarantee on ROI violation, we employ the following decomposition, which decouples the anytime violation into oracle estimation error terms and estimated violation.

$$\mathbb{E}\left[\sum_{s=1}^{t} \gamma c(v_s, b_s) - r(v_s, b_s)\right] \leqslant \mathbb{E}\left[\sum_{s=1}^{t} (\mathbb{E}_v[\gamma c(v, b_s) - r(v, b_s)])^+\right]$$

$$\leqslant \mathbb{E}\left[\sum_{s=1}^{t} (\mathbb{E}_v[\gamma \check{c}_s(v, b_s) - \hat{r}_s(v, b_s)])^+\right] + \mathcal{E}_r(T, p) + \gamma \mathcal{E}_c(T, p)$$

$$\leqslant \mathbb{E}[Q_\gamma(t)]/\eta_\gamma + \mathcal{E}_r(T, p) + \gamma \mathcal{E}_c(T, p).$$

Finally, applying Lemma 2 to bound the expected queue lengths yields the violation bound in Theorem 1. Then, we provide the proof of regret bound.

**Regret (before stopping and after stopping):** We begin by introducing the decomposition of regret in (4), which separates the decision process into the reward gap before stopping and after stopping.

$$\text{Regret}(T) = \mathbb{E}\left[\sum_{t=1}^{T} r(v_t, b^*) - \sum_{t=1}^{\tau} r(v_t, b_t)\right] \leqslant \underbrace{\nu^* \mathbb{E}[T - \tau]}_{\text{Regret after stopping}} + \underbrace{\mathbb{E}\left[\sum_{t=1}^{\tau} (r(v_t, b^*) - r(v_t, b_t))\right]}_{\text{Regret before stopping}}$$

For *regret before stopping*, we apply Lemma 1 with $b = b^*$, rearrange the terms, sum the resulting inequality up to the stopping time $\tau \leqslant T$, and take expectations, which yields

$$\mathbb{E}\left[\sum_{t=1}^{\tau} (r(v_t, b^*) - r(v_t, b_t))\right] \leqslant 4 \sum_{t=1}^{\tau} \varepsilon_t^r + \mathbb{E}\left[\sum_{t=1}^{\tau} \Phi'(Q_\rho(t, b^*))\eta_\rho(\mathbb{E}_v[\check{c}_t(v, b^*)] - \rho)^+\right]$$

$$+ \mathbb{E}\left[\sum_{t=1}^{\tau} \Phi'(Q_\gamma(t, b^*))\eta_\gamma(\mathbb{E}_v[\gamma \check{c}_t(v, b^*) - \hat{r}_t(v, b^*)])^+\right]$$

The following lemma, which provides an upper bound on the cross terms, completes the proof of the regret bound after stopping.

**Lemma 3** *Let b\* be the optimal solution to (2), then the following results hold that,*

$$\mathbb{E}\left[\sum_{t=1}^{\tau} \Phi'(Q_\rho(t, b^*))\eta_\rho(\mathbb{E}_v[\check{c}_t(v, b^*)] - \rho)^+\right] \leqslant 2\eta_\rho C_{max}^2/T^2 = O(1),$$

$$\mathbb{E}\left[\sum_{t=1}^{\tau} \Phi'(Q_\gamma(t, b^*))\eta_\gamma(\mathbb{E}_v[\gamma\check{c}_t(v, b^*) - \hat{r}_t(v, b^*)])^+\right] \leqslant 2\eta_\gamma\gamma C_{\max}^2/T^2 = O(1).$$

For *regret after stopping*, we first recall the definition of stopping time $\tau$ that is $\tau = \operatorname*{argmin}_{\tau' \in [T]}\left\{\tau' \mid \sum_{t=1}^{\tau'} c(v_t, b_t) \geqslant B\right\}$, which gives

$$\sum_{t=1}^{\tau}(\check{c}_t(v_t, b_t) - \rho) + \tau\rho + \sum_{t=1}^{\tau}(c_t(v_t, b_t) - \check{c}_t(v_t, b_t)) \geqslant B,$$

Taking the expectation and rearranging the inequality, we have

$$\mathbb{E}[T - \tau] = \frac{1}{\rho}\mathbb{E}[B - \tau\rho] \leqslant \frac{1}{\rho}\mathbb{E}\left[\sum_{t=1}^{\tau}(\check{c}_t(v_t, b_t) - \rho) + \sum_{t=1}^{\tau}(c_t(v_t, b_t) - \check{c}_t(v_t, b_t))\right]$$

$$\leqslant \frac{1}{\rho}\mathbb{E}\left[\sum_{t=1}^{\tau}(\mathbb{E}_v[\check{c}_t(v, b_t)] - \rho)\right] + \mathcal{E}_c(T, p)$$

Combining with the fact in (11), we complete the proof of Theorem 1.

## 5 INSTANTIATIONS OF THE MODULAR ALGORITHM: SETTINGS, RESULTS, AND COMPARISONS

In this section, we instantiate L2FOB in several online bidding environments and learning models, specify the oracle choices and their cumulative errors, and derive regret and violation bounds via Theorem 1. For first-price auctions in Wang et al. (2023), using the same empirical estimators, we obtain a regret bound $\tilde{O}\big((1 + \nu^*/\rho)\sqrt{T}\big)$, improving their result by a factor of $1/\rho$. Note that Wang et al. (2023) does not incorporate ROI constraints. For second-price auctions under the uncertainty problem in Castiglioni et al. (2025), while they do not provide an ROI violation metric, instantiating their estimators within L2FOB yields sublinear regret and sublinear violation, and our analysis explicitly considers hard-stopping in the regret, which theirs does not. For constrained contextual bandits Guo et al. (2025), which deliver state-of-the-art guarantees without Slater's condition, adopting the same learning oracles allows L2FOB to match their guarantees. Moreover, when stronger estimators are available, our framework can provide refined regret and violation bounds, whereas their algorithm cannot. Note that across all these settings, our results provide an *anytime* guarantee on ROI violation, whereas prior studies state guarantees only at the terminal horizon $T$.

### 5.1 FIRST-PRICE AUCTION UNDER BUDGET CONSTRAINTS

We consider the same setting as Wang et al. (2023), where a single advertiser participates in repeated first-price auctions over $T$ rounds with a total spend budget $B = T\rho$. In round $t$, the bidder has private value $v_t \in [0, v_{max}]$, submits a bid $b_t \in \mathbb{R}^+$, faces the highest competing bid $d_t$, and wins iff $b_t \geqslant d_t$. Let $x_t := \mathbb{I}\{b_t \geqslant d_t\}$. Then the reward and cost functions are

$$r_t(v_t, b_t) = x_t(v_t - b_t), \ c_t(v_t, b_t) = x_t b_t,$$

and the process stops when the budget is exhausted (or at $T$), i.e., feasibility requires $\sum_{t=1}^{\tau} c_t(v_t, b_t) \leqslant B$ for the stopping time $\tau \leqslant T$. Equivalently, the expected reward and cost (conditioning on $d_t$ only) can be written via the win probability $G(b) := \Pr(d_t \leqslant b)$ as

$$r(v, b) = (v - b)G(b), \ c(v, b) = bG(b).$$

In Wang et al. (2023), the first-price-with-budget models inherently satisfy Assumption 1. When the platform reveals $d_s$ after each round, estimate the win CDF by the empirical CDF $\hat{G}_t(b) =$

$\frac{1}{t-1} \sum_{s=1}^{t-1} \mathbb{I}\{d_s \leqslant b\}$ and use

$$\bar{r}_t(v, b) = (v - b)\widehat{G}_t(b), \ \bar{c}_t(v, b) = b\widehat{G}_t(b).$$

Uniform concentration then yields high-probability oracle error bounds $\varepsilon_t^r, \ \varepsilon_t^c$, ensuring that Assumption 1 holds. Specifically, we have

$$\mathcal{E}_r(T, p) = O\Big(\sqrt{T \ln \tfrac{T}{p}}\Big), \ \mathcal{E}_c(T, p) = O\Big(\sqrt{T \ln \tfrac{T}{p}}\Big),$$

Plugging the above cumulative learning error into Theorem 1, L2FOB achieves $\tilde{O}\big((1 + \nu^*/\rho)\sqrt{T}\big)$ regret and $\tilde{O}(\sqrt{T})$ anytime ROI violation, where $\tilde{O}(\cdot)$ hides logarithmic factors in $T$.

**Remark 2** *Our guarantee substantially improves over Wang et al. (2023). Although they report a $\tilde{O}(\sqrt{T})$ regret, the dependences on $\rho$ and $\nu^*$ are hidden, and the exact bound is $\tilde{O}\big((1+\nu^*/\rho^2)\sqrt{T}\big)$ (see Appendix B.3 of Wang et al. (2023)). Consequently, their result is meaningful only in the large budget regime $B = T\rho = \Theta(T)$, since $\nu^*/\rho^2$ grows without bound as the budget shrinks. In contrast, our bound scales as $\tilde{O}\big((1 + \nu^*/\rho)\sqrt{T}\big)$ and remains informative under small budgets $B = \Omega(\sqrt{T})$. Moreover, Wang et al. (2023) only considers budget constraints, whereas L2FOB handles both budget and ROI constraints and provides explicit guarantees on regret and on the cumulative amount of violation. For budget constraints, Slater's condition typically holds because the bidder can choose a null (zero-spend) action; nevertheless, L2FOB does not require it, enabling harder settings in which every bid incurs a positive cost or a minimum spend is mandated.*

## 5.2 BIDDING UNDER UNCERTAINTY WITH BUDGET AND ROI CONSTRAINTS

We adopt the setting of Castiglioni et al. (2025), which emphasizes uncertainty in online bidding: the revenue and cost of advertising campaigns are unknown and must be learned online from sequential data. An advertiser manages $N$ subcampaigns over $T$ rounds. In round $t$, a bid $x_{j,t} \in X_j \subset \mathbb{R}^+$ is chosen for each subcampaign $C_j$, yielding expected clicks $n_j(x_{j,t})$ and expected cost $c_j(x_{j,t})$. With per–click values $v_j$, the round-$t$ expected revenue is $\sum_{j=1}^N v_j n_j(x_{j,t})$. Then the per-round optimization problem is

$$\max_{(x_{j,t})_{j=1}^N} \sum_{j=1}^N v_j n_j(x_{j,t}) \text{ s.t. } \frac{\sum_{j=1}^N v_j n_j(x_{j,t})}{\sum_{j=1}^N c_j(x_{j,t})} \geqslant \gamma, \ \sum_{j=1}^N c_j(x_{j,t}) \leqslant \rho.$$

Within a round, each subcampaign's bid is fixed, but bids may be updated across rounds. Set the action to be the vector $b_t := x_t$ and define

$$r(v_t, b_t) = \sum_{j=1}^N v_j n_j(b_{t,j}), \ c(v_t, b_t) = \sum_{j=1}^N c_j(b_{t,j}).$$

This embeds the multi–subcampaign problem directly into our constrained online learning formulation, and it satisfies our regression-oracle assumption by plugging in the estimators for $n_j(\cdot)$ and $c_j(\cdot)$. Following Castiglioni et al. (2025), model $n_j(\cdot)$ and $c_j(\cdot)$ with independent Gaussian processes and use UCB/LCB envelopes. Let $\sigma_{j,t-1}^{(n)}(x)$ and $\sigma_{j,t-1}^{(c)}(x)$ be posterior standard deviations and $\gamma_t$ the usual GP-UCB confidence schedule. Then the oracle errors can be taken as

$$\varepsilon_t^r = \Theta\big(\sqrt{\gamma_t}\sigma_{j_t,t-1}^{(n)}(x_{j_t,t})\big), \ \varepsilon_t^c = \Theta\big(\sqrt{\gamma_t}\sigma_{j_t,t-1}^{(c)}(x_{j_t,t})\big).$$

Then, by standard Gaussian-process concentration results Srinivas et al. (2009); Krause & Ong (2011); Kim & Sanz-Alonso (2025) and following the derivation in Castiglioni et al. (2025), we obtain

$$\mathcal{E}_r(T, p) = \tilde{O}\Big(\sqrt{T \sum_{j=1}^N \gamma_{j,T}}\Big), \ \mathcal{E}_c(T, p) = \tilde{O}\Big(\sqrt{T \sum_{j=1}^N \gamma_{j,T}}\Big),$$

where $\gamma_{j,T}$ is the GP information gain for subcampaign $j$. Instantiating L2FOB with the GP-UCB oracles and applying Theorem 1 yields

$$\text{Regret}(T) = \tilde{O}\Big(\big(1 + \nu^*/\rho\big)\sqrt{T \sum_{j=1}^N \gamma_{j,T}}\Big), \ \text{V}_{\text{ROI}}(t) = \tilde{O}\Big(\sqrt{T \sum_{j=1}^N \gamma_{j,T}}\Big), \ \forall t \in [T].$$

**Remark 3** *In this setting, L2FOB is, to the best of our knowledge, the first method to attain optimal-order regret together with ROI violation guarantees. By contrast, Castiglioni et al. (2025) quantifies safety by the number of violating rounds. Their analysis implies that achieving sublinear regret entails $\Theta(T)$ violating rounds under this count-based metric, and their regret analysis does not account for budget-induced hard-stopping. Counting violations is improper for online bidding, where pacing can recycle unspent budget and small, controlled per-round violations can be offset later. L2FOB instead measures and controls the cumulative ROI violation and explicitly models the hard-stopping induced by the budget, yielding guarantees that are stronger in theory and better aligned with practice than Castiglioni et al. (2025). Moreover, L2FOB can leverage recent advances in Bayesian optimization analysis Iwazaki & Takeno (2025) to tighten oracle errors, further improving regret and violation bounds.*

### 5.3 CONSTRAINED CONTEXTUAL BANDITS FOR BIDDING

We next study the constrained contextual bandit formulation for online bidding in Guo et al. (2025). Contextual bandits fit our framework naturally: the bid can be modeled as a discrete action $b_t \in \mathcal{A}$ or as a policy mapping contexts to actions. The reward and cost are unknown, context-dependent functions that must be learned online from noisy bandit feedback. They made a similar assumption on learning oracle errors, which gives $\mathcal{E}_r(T, p) = \tilde{O}(\sqrt{T})$, $\mathcal{E}_c(T, p) = \tilde{O}(\sqrt{T})$. Instantiating L2FOB with these oracles and applying Theorem 1 gives

$$\text{Regret}(T) = \tilde{O}\big((1 + \nu^*/\rho)\sqrt{T}\big), \ \text{V}_{\text{ROI}}(t) = \tilde{O}(\sqrt{T}), \ \forall t \in [T].$$

Our results match the best-known guarantees for constrained contextual bandits, as established by Guo et al. (2025). Their approach also requires prior knowledge of the context distribution to enforce constraints in a mean-field manner. However, although Guo et al. (2025) considers general constraints, it does not explicitly handle multiple simultaneous constraints, so its applicability to online bidding with both budget and ROI constraints is unclear. L2FOB offers *anytime* control of violations, thereby covering the metric $\text{V}_{\text{ROI}}(T)$ in Guo et al. (2025). Moreover, our results are modular and adaptive, depending explicitly on oracle errors, which enables refined guarantees as highlighted in the remark below.

**Remark 4 (Bridging the gap.)** *The regret lower bound for non-contextual bandits with budget constraints is proved to be $\Omega\big(\sqrt{T\nu^*} + (\nu^*/\rho)\sqrt{B}\big)$ in Badanidiyuru et al. (2018). For contextual bandits with budget constraints, the best-known guarantee is $\tilde{O}\big((1 + \nu^*/\rho)\sqrt{T}\big)$ as shown in Guo et al. (2025), which leaves a gap between contextual and non-contextual settings. If one could design learning oracles with $\mathcal{E}_r(T, p) = \tilde{O}\big(\sqrt{T\nu^*}\big)$, $\mathcal{E}_c(T, p) = \tilde{O}\big(\sqrt{B}\big)$, then Theorem 1 would yield $\text{Regret}(T) = \tilde{O}\big(\sqrt{T\nu^*} + (\nu^*/\rho)\sqrt{B}\big)$, which matches the non-contextual lower bound. This hypothesis is plausible because these error bounds hold in the non-contextual case Badanidiyuru et al. (2018). At present, it is unclear how to attain such adaptive learning oracles in contextual settings. Even if such oracles were available, the methods of Guo & Liu (2025); Guo et al. (2025) would still not reach this bound because their dual updates made it hard to introduce a flexible choice of $\eta_\rho$, which our look-ahead design enables.*

## 6 CONCLUSION

In this paper, we study safe online bidding under budget and return-on-investment (ROI) constraints, where the budget imposes a hard stopping condition and the ROI enforces a long-run profitability threshold. We introduced L2FOB, a Look-ahead Lyapunov Framework for Online Bidding that pairs optimistic/pessimistic estimators with look-ahead virtual queue penalties and yields adaptive bounds on regret and cumulative violation without requiring Slater's condition. Our algorithm provides a modular, safe, and optimal framework. We instantiate L2FOB in several online bidding environments and show that the resulting guarantees match or improve the best-known results. The experiments further validate the theoretical results.

## ACKNOWLEDGMENTS

This work was supported by the National Natural Science Foundation of China under Grant 62302305.

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

## A    RELATED WORKS

Learning in repeated auctions with constraints has been studied across auction formats, feedback regimes, and safety notions. For first-price auctions with a single advertiser and a hard budget, Wang et al. (2023) propose primal–dual methods with $\tilde{O}(\sqrt{T})$ regret under both full-information and one-sided feedback; In Tang et al. (2014); Balseiro et al. (2020; 2023), second-price auctions with budget constraints is also considered, where Tang et al. (2014) claims their results holds under large budget regime ; Badanidiyuru et al. (2023) study first-price auction while focusing on the linear model setting; with only binary win/loss signals, Balseiro et al. (2019) employ cross-learning to obtain $\tilde{O}(T^{2/3})$ regret, and Zhang et al. (2022) exploit side information "hints" about the maximum competing bid. In Han et al. (2025), they leverage graph-feedback and partial-order structure in first-price markets to attain optimal regret rates. Feng et al. (2023); Golrezaei et al. (2021a) studied second-price auctions with budget and return-on-spend (RoS) constraints and Vaze & Sinha (2025) considered the case with only RoS constraints. Castiglioni et al. (2025) study budget and ROI constrained bidding under full uncertainty via Bayesian optimization. Their safety metric counts violating rounds rather than measuring cumulative violations. Castiglioni et al. (2024) studies a general online learning problem with both budget and ROI constraints, but in a setting without context. They do not require prior knowledge of the Slater constant, although they still assume that a Slater condition holds. Vijayan et al. (2025) also adopts UCB/LCB style estimators to design algorithms and establishes $\tilde{O}(\sqrt{T})$ regret and violation bounds under budget and RoS constraints. However, they do not consider hard stopping upon budget exhaustion and focus on a bandit-only setting. To avoid assuming Slater's condition, their algorithm solves a linear program each round, which introduces additional computational overhead. There are also works on multi-agent bidding dynamics Balseiro & Gur (2019); Lucier et al. (2024), which analyze equilibria and regret; this line differs from our single-advertiser focus. Despite this progress, there is no single framework that uniformly handles general reward/cost models, heterogeneous feedback, and *simultaneous* budget and ROI constraints with amount-based safety guarantees. Moreover, analyses focused on budget constraints often suppress the dependence on the average spend rate $\rho = B/T$, obscuring behavior in small-budget regimes common in practice.

Another parallel line of work studies contextual bandits (CB) with constraints. Early CB algorithms relied on classification oracles Dudik et al. (2011); Agarwal et al. (2014), whereas later work developed regression oracle methods Foster et al. (2018); Foster & Rakhlin (2020); Simchi-Levi & Xu (2022) that are more computationally practical and broadly applicable. Our oracle assumption subsumes these regression-based designs but is more general: we allow arbitrary function classes and plug-in estimators (including, but not limited to, contextual bandit regressors). Within constrained CB, much attention has focused on knapsack (budget) constraints Badanidiyuru et al. (2014); Agrawal & Devanur (2014); Wu et al. (2015); Agrawal & Devanur (2016); Badanidiyuru et al. (2018); Sivakumar et al. (2022); Chzhen et al. (2024); Guo & Liu (2025). For linear models under Slater's condition, optimal guarantees of $\tilde{O}((1 + \nu^*/b)\sqrt{T})$ have been established Agrawal & Devanur (2016) and be further extended to general function class in Guo & Liu (2025), but these results address only budget control, whereas practical bidding must also enforce profitability (ROI). Long-run constraints with cumulative-violation metrics were studied in Slivkins et al. (2023), which achieved $\tilde{O}(\sqrt{T}/\delta)$ regret and violation under Slater's condition. Subsequent work Guo & Liu (2024); Guo et al. (2025) tried to relax Slater, where Guo et al. (2025) obtained $\tilde{O}((1+\nu^*/b)\sqrt{T})$ regret for knapsack and $\tilde{O}(\sqrt{T})$ regret and violation for certain long-term constraints without Slater's condition. However, it neither offers modular, adaptive guarantees nor provides a single mechanism that simultaneously handles budget and general long-term constraints. In contrast, our framework unifies budget and ROI control, eliminates the need for Slater's condition, and remains compatible with general regression oracles, yielding adaptive guarantees across modeling choices.

## B    DISCUSSION ON LYAPUNOV STABILITY

Lyapunov drift analysis has been widely used to study the stability of control policies in stochastic queueing networks (see Hajek (1982); Tassiulas & Ephremides (1992)). A policy is called stable when the induced queue lengths remain finite or uniformly bounded, and smaller queues typically indicate better performance. Modern analytical frameworks providing upper bounds on queue lengths to ensure stability are developed in Bertsimas et al. (2001); Eryilmaz & Srikant (2012). For a broader

introduction to Lyapunov optimization and its applications, we refer readers to Neely (2022); Srikant & Ying (2014).

## C COMPARISON TABLE

In this section, we provide a detailed comparison of settings in Section 5.

| Reference | Budget constraint | ROI constraint | Regret | ROI violation | Budget regime | Slater's condition free | Generalization |
|---|---|---|---|---|---|---|---|
| Wang et al. (2023) | ✓ | ✗ | $\tilde{\mathcal{O}}\left((1+\frac{\nu^*}{\rho^2})\sqrt{T}\right)$ | ✗ | $\Omega(T)$ | ✗ | ✗ |
| This work | ✓ | ✓ | $\tilde{\mathcal{O}}\left((1+\frac{\nu^*}{\rho})\sqrt{T}\right)$ | $\tilde{\mathcal{O}}(\sqrt{T})$ | $\Omega(\sqrt{T})$ | ✓ | ✓ |

Table 1: Our results and related work in the first-price auction.

| Reference | Budget constraint | ROI constraint | Regret | ROI violation | Budget regime | Slater's condition free | Generalization |
|---|---|---|---|---|---|---|---|
| Castiglioni et al. (2025) | ✓ | ✓ | $\tilde{\mathcal{O}}\left(\sqrt{T\gamma_{1:N}^T}\right)$ | ✗ | ✗ | ✓ | ✗ |
| This work | ✓ | ✓ | $\tilde{\mathcal{O}}\left((1+\frac{\nu^*}{\rho})\sqrt{T\gamma_{1:N}^T}\right)$ | $\tilde{\mathcal{O}}(\sqrt{T\gamma_{1:N}^T})$ | $\Omega(\sqrt{T})$ | ✓ | ✓ |

Table 2: Our results and related work in online bidding under uncertainty, where $\gamma_{1:N}^T := \sum_{j=1}^N \gamma_{j,T}$.

| Reference | Budget constraint | ROI constraint | Regret | ROI violation | Budget regime | Slater's condition free | Generalization |
|---|---|---|---|---|---|---|---|
| Guo et al. (2025) | ✓ | ✗ | $\tilde{\mathcal{O}}\left((1+\frac{\nu^*}{\rho})\sqrt{T}\right)$ | ✗ | $\Omega(\sqrt{T})$ | ✓ | ✗ |
| This work | ✓ | ✓ | $\tilde{\mathcal{O}}\left((1+\frac{\nu^*}{\rho})\sqrt{T}\right)$ | $\tilde{\mathcal{O}}(\sqrt{T})$ | $\Omega(\sqrt{T})$ | ✓ | ✓ |

Table 3: Our results and related work in the contextual bandit.

## D EXPERIMENTS

In this section, we evaluate the proposed algorithm across different auction settings and learning methods. Specifically, we first reproduce the first-price auction environment of Wang et al. (2023); we then consider the constrained bandit setting of Guo et al. (2025), where the reward is modeled as advertisement relevance; we also conduct a sensitivity analysis on the choice of $\eta_\gamma$ in the first-price auction environment.

For each experiment, we report the average reward $\frac{1}{t}\sum_{s=1}^t r_s$ and the real-time ROI $\frac{\sum_{s=1}^t r_s}{\sum_{s=1}^t c_s}$. Each curve shows the mean over 20 independent runs with different random seeds, and shaded regions indicate 95% confidence intervals.

While L2FOB theoretically assumes access to the context distribution to perform mean-field evaluation of the constraints, we deploy a variant that directly utilizes the current context $v_t$ when selecting $b_t$ in practice. Under this implementation, L2FOB still achieves superior performance in all experiments, demonstrating strong robustness.

### D.1 FIRST-PRICE AUCTION UNDER BOTH BUDGET AND ROI CONSTRAINTS

In this setting, we follow the empirical setup of Wang et al. (2023) and additionally impose a return-on-investment (ROI) constraint to evaluate the algorithm's profitability. Specifically, the time horizon is set as $T = 10^4$ with a total budget $B = 10^2$, yielding a per-round budget $\rho = B/T = 0.01$.

The ROI threshold is $\gamma = 1.8$. The private value (context) is sampled as $v_t \sim \mathcal{N}(0.6, 0.1)$, and the competing bid as $d_t \sim \mathcal{N}(0.4, 0.1)$. We restrict $v_t$ to the interval $[0, 1]$, and any values of $v_t$ or $d_t$ exceeding 1 are truncated to 1.

For the baseline of Wang et al. (2023), we set the step size to $\epsilon = 1/\sqrt{T}$, as specified in their paper. For L2FOB, we use constant learning rates for the dual variables, with $\eta_\rho = \eta_\gamma = 0.6$. This choice does not conflict with our setup in L2FOB. As noted after Theorem 1, our framework allows $\eta_\rho$ and $\eta_\gamma$ beyond the $\Theta(\sqrt{T})$ scaling. When the oracle errors are of order $\widetilde{O}(\sqrt{T})$, constant order choices of $\eta_\rho$ and $\eta_\gamma$ also suffice. Intuitively, $\eta_{\{\rho,\gamma\}}$ governs the trade-off between reward and the budget/ROI constraints. Thanks to our rectified design that incorporates the clip operator $(\cdot)^+$, even relatively large values of $\eta_{\{\rho,\gamma\}}$ have a limited adverse impact on performance; see the sensitivity study in Appendix D.3.

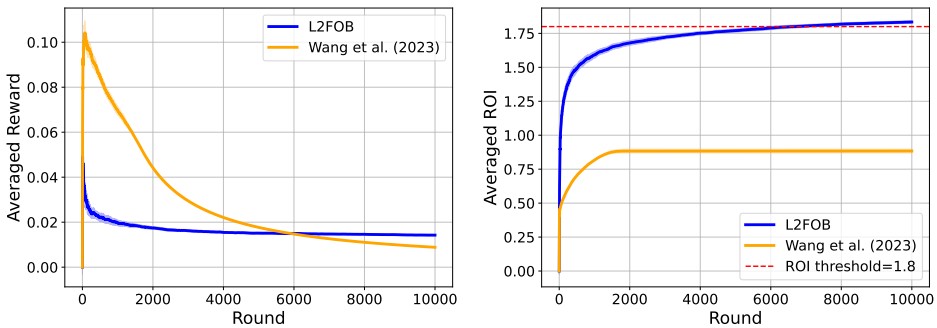

Figure 1: The averaged reward and averaged ROI of L2FOB and algorithm in Wang et al. (2023) in first-price auction setting.

The results in Figure 1 show that L2FOB significantly outperforms the algorithm of Wang et al. (2023) in both reward and ROI under the first-price auction setting. L2FOB satisfies the ROI threshold stably over time, consistent with our *anytime* ROI violation guarantee (Theorem 1). By contrast, the method of Wang et al. (2023), which does not explicitly enforce an ROI constraint, fails to reach the target profitability level and maintains a persistently low ROI. Moreover, L2FOB achieves higher and more stable average rewards throughout the horizon.

## D.2    CONSTRAINED CONTEXTUAL BANDIT FOR BIDDING

We also consider the constrained contextual bandit setting in Guo et al. (2025). In this setting, we conduct experiments on the large-scale learning-to-rank dataset MSLR-WEB30k Qin & Liu (2013), where each bidding action yields a corresponding relevance score. Following the experimental configuration of Guo et al. (2025), we set $\gamma = 1.3$, $T = 5000$, and $B = 1000$, yielding a per-round budget of $\rho = B/T = 0.2$. Within the bandit abstraction, the action (bid) $b_t$ corresponds to selecting an arm. We take the contextual feature $v_t$ and the reward $r_t$ from the dataset, and sample the cost as $c_t \sim \mathcal{N}(0.2, 0.02)$. To better approximate real-world noise, we corrupt both the observed rewards and costs by adding Gaussian noise $\mathcal{N}(0, 0.05)$.

As in Guo et al. (2025), we use a gradient-boosted tree estimator as the learning oracle for the reward model and estimate the cost via empirical means. For the scaling terms, we also set $\eta_\rho = \eta_\gamma = 0.6$ in our algorithm. Because Guo et al. (2025) considers a single-constraint setting, we adapt their method to a multiple-constraint setting that simultaneously enforces both a budget constraint and an ROI constraint. The experimental results are reported in Figure 2. While our adaptation enables the algorithm of Guo et al. (2025) to operate under both budget and ROI constraints, the results show that our algorithm consistently outperforms Guo et al. (2025) in both average reward and ROI. Moreover, L2FOB maintains stable ROI performance over the time horizon and ultimately satisfies the ROI target.

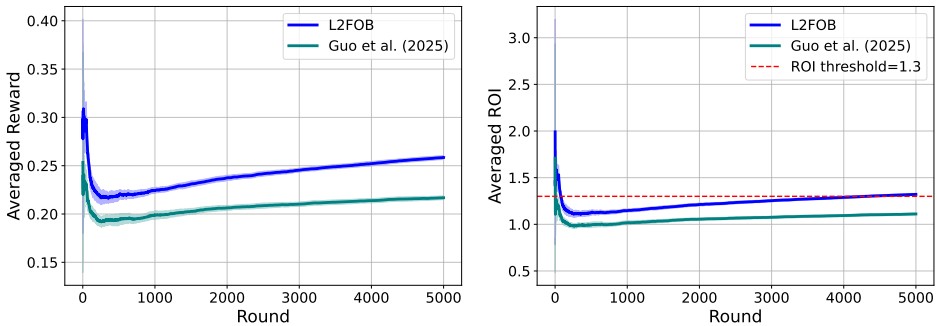

Figure 2: The averaged reward and averaged ROI of L2FOB and algorithm in Guo et al. (2025) in the constrained contextual bandit setting.

### D.3 SENSITIVITY ANALYSIS

Recall that in L2FOB we set $\eta_\rho = \eta_\gamma = \sqrt{T}$. However, L2FOB allows a more flexible choice of these scaling terms, both theoretically and empirically. Specifically, $\eta_{\{\rho,\gamma\}}$ may range from constant order $\Theta(1)$ up to $\Theta(T)$ without affecting our theoretical guarantees. This is due to our rectified design that incorporates the clipping operator $(\cdot)^+$ on the constraints, which allows the algorithm to focus on optimizing reward for actions predicted to be safe. Consequently, once the scaling terms are sufficiently large, further increases have little impact on performance. We focus on studying the influence of ROI scaling terms $\eta_\gamma$ and evaluate four representative values, that is $\eta_\gamma \in \{0.06, 0.6, 6, 60\}$.

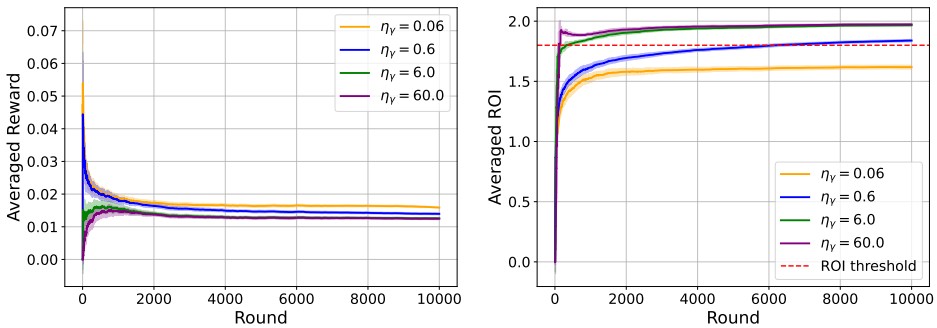

Figure 3: Sensitivity study on scaling term $\eta_\gamma$.

Figure 3 aligns with our theoretical analysis of the scaling terms. Specifically, setting $\eta_\gamma \in \{0.06, 0.6, 6\}$ strikes a mild tradeoff between cumulative reward and ROI violation. However, when we set $\eta_\gamma = 60$, the performance differs only slightly from $\eta_\gamma = 6$. This aligns with our theoretical analysis: thanks to the rectified design of L2FOB, the algorithm does not become over-conservative under large constraint scaling, allowing more flexible control.

## E DETAILED PROOFS

### E.1 PROOF OF LEMMA 1

**Lemma 4 (Restatement of Lemma 1)** *Let $b_t$ be the bidding action selected by the L2FOB algorithm at round $t$. Then, for any bidding action $b \in \mathbb{R}^+$, the following inequality holds:*

$$\hat{r}_t(v_t, b) - \hat{r}_t(v_t, b_t) + \Phi(Q_\rho(t)) - \Phi(Q_\rho(t-1)) + \Phi(Q_\gamma(t)) - \Phi(Q_\gamma(t-1))$$
$$\leqslant 4\varepsilon_t^r + \Phi'(Q_\rho(t,b))\eta_\rho(\mathbb{E}_v[\check{c}_t(v,b)] - \rho)^+ + \Phi'(Q_\gamma(t,b))\eta_\gamma(\mathbb{E}_v[\gamma\check{c}_t(v,b) - \hat{r}_t(v,b)])^+,$$

To give the proof, we first give the following lemma that bridges the decision process with the potential drift:

**Lemma 5** *Let $\Phi(\cdot)$ be any convex potential function, and update $Q_\rho(t)$ and $Q_\gamma(t)$ according to (9) and (10). Then, for every round $t$,*

$$\Phi\big(Q_\rho(t)\big) \leqslant \Phi\big(Q_\rho(t-1)\big) + \Phi'(Q_\rho(t))\eta_\rho(\mathbb{E}_v[\check{c}_t(v, b_t)] - \rho)^+,$$

$$\Phi\big(Q_\gamma(t)\big) \leqslant \Phi\big(Q_\gamma(t-1)\big) + \Phi'(Q_\gamma(t))\eta_\gamma(\mathbb{E}_v[\gamma\check{c}_t(v, b_t) - \hat{r}_t(v, b_t)])^+.$$

**Proof 1** *We prove the first inequality, then the second follows by the same argument. By convexity (and differentiability) of $\Phi$, for any $x, y$ we have $\Phi(y) \leqslant \Phi(x) + \Phi'(y)(y - x)$. With $y = Q_\rho(t)$ and $x = Q_\rho(t-1)$,*

$$\Phi\big(Q_\rho(t)\big) \leqslant \Phi\big(Q_\rho(t-1)\big) + \Phi'\big(Q_\rho(t)\big)\big(Q_\rho(t) - Q_\rho(t-1)\big)$$
$$= \Phi\big(Q_\rho(t-1)\big) + \Phi'\big(Q_\rho(t)\big)\eta_\rho(\mathbb{E}_v[\check{c}_t(v, b_t)] - \rho)^+,$$

*where the last equality holds due to the update rule in (9).*

From the decision rule in (8), we can directly obtain that for any bid decision $b$, the following inequality holds:

$$\hat{r}_t(v_t, b) - \Phi'(Q_\rho(t, b))\eta_\rho(\mathbb{E}_v[\check{c}_t(v, b)] - \rho)^+ - \Phi'(Q_\gamma(t, b))\eta_\gamma(\mathbb{E}_v[\gamma\check{c}_t(v, b) - \hat{r}_t(v, b)])^+$$
$$\leqslant \hat{r}_t(v_t, b_t) - \Phi'(Q_\rho(t))\eta_\rho(\mathbb{E}_v[\check{c}_t(v, b_t)] - \rho)^+ - \Phi'(Q_\gamma(t))\eta_\gamma(\mathbb{E}_v[\gamma\check{c}_t(v, b_t) - \hat{r}_t(v, b_t)])^+,$$

where we have $Q_\rho(t) = Q_\rho(t, b_t)$ and $Q_\gamma(t) = Q_\gamma(t, b_t)$ according to the definition. Rearrange the inequality, we can get:

$$\hat{r}_t(v_t, b) - \hat{r}_t(v_t, b_t) + \Phi'(Q_\rho(t))\eta_\rho(\mathbb{E}_v[\check{c}_t(v_t, b_t)] - \rho)^+ + \Phi'(Q_\gamma(t))\eta_\gamma(\mathbb{E}_v[\gamma\check{c}_t(v_t, b_t) - \hat{r}_t(v_t, b_t)])^+$$
$$\leqslant \Phi'(Q_\rho(t, b))\eta_\rho(\mathbb{E}_v[\check{c}_t(v_t, b)] - \rho)^+ + \Phi'(Q_\gamma(t, b))\eta_\gamma(\mathbb{E}_v[\gamma\check{c}_t(v_t, b) - \hat{r}_t(v_t, b)])^+.$$

We can then apply Lemma 5 to get:

$$\hat{r}_t(v_t, b) - \hat{r}_t(v_t, b_t) + \Phi(Q_\rho(t)) - \Phi(Q_\rho(t-1)) + \Phi(Q_\gamma(t)) - \Phi(Q_\gamma(t-1))$$
$$\leqslant \Phi'(Q_\rho(t, b))\eta_\rho(\mathbb{E}_v[\check{c}_t(v_t, b)] - \rho)^+ + \Phi'(Q_\gamma(t, b))\eta_\gamma(\mathbb{E}_v[\gamma\check{c}_t(v_t, b) - \hat{r}_t(v_t, b)])^+.$$

Add $r(v_t, b) - r(v_t, b_t)$ on both sides and rearrange the inequality, we can get:

$$r(v_t, b) - r(v_t, b_t) + \Phi(Q_\rho(t)) - \Phi(Q_\rho(t-1)) + \Phi(Q_\gamma(t)) - \Phi(Q_\gamma(t-1))$$
$$\leqslant r(v_t, b) - \hat{r}_t(v_t, b) + \hat{r}_t(v_t, b_t) - r(v_t, b_t)$$
$$\quad + \Phi'(Q_\rho(t, b))\eta_\rho(\mathbb{E}_v[\check{c}_t(v_t, b)] - \rho)^+ + \Phi'(Q_\gamma(t, b))\eta_\gamma(\mathbb{E}_v[\gamma\check{c}_t(v_t, b) - \hat{r}_t(v_t, b)])^+$$
$$\leqslant |r(v_t, b) - \hat{r}_t(v_t, b)| + |\hat{r}_t(v_t, b_t) - r(v_t, b_t)|$$
$$\quad + \Phi'(Q_\rho(t, b))\eta_\rho(\mathbb{E}_v[\check{c}_t(v_t, b)] - \rho)^+ + \Phi'(Q_\gamma(t, b))\eta_\gamma(\mathbb{E}_v[\gamma\check{c}_t(v_t, b) - \hat{r}_t(v_t, b)])^+$$
$$\leqslant 4\varepsilon_t^r + \Phi'(Q_\rho(t, b))\eta_\rho(\mathbb{E}_v[\check{c}_t(v_t, b)] - \rho)^+ + \Phi'(Q_\gamma(t, b))\eta_\gamma(\mathbb{E}_v[\gamma\check{c}_t(v_t, b) - \hat{r}_t(v_t, b)])^+,$$

where the last inequality holds from the online regression oracle assumption and the optimistic reward estimators, thereby completing the proof.

### E.2 PROOF OF LEMMA 3

**Lemma 6 (Restatement of Lemma 3)** *Let $b^*$ be an optimal solution to (2). Then, with stopping time $\tau$,*

$$\mathbb{E}\left[\sum_{t=1}^{\tau} \Phi'\big(Q_\rho(t, b^*)\big)\eta_\rho(\mathbb{E}_v[\check{c}_t(v, b^*)] - \rho)^+\right] \leqslant 2\eta_\rho C_{\max}^2/T^2 = O(1),$$

$$\mathbb{E}\left[\sum_{t=1}^{\tau} \Phi'\big(Q_\gamma(t, b^*)\big)\eta_\gamma(\mathbb{E}_v[\gamma\check{c}_t(v, b^*) - \hat{r}_t(v, b^*)])^+\right] \leqslant 2\eta_\gamma\gamma C_{\max}^2/T^2 = O(1).$$

We prove the first result, then the second follows by the same argument. On the high-probability event $\omega$ from Assumption 1, we have, for all $t \in [T]$ and all $(v,b) \in \mathcal{X} \times \mathbb{R}^+$,

$$\hat{r}_t(v,b) \geqslant r(v,b), \ c(v,b) \geqslant \check{c}_t(v,b).$$

Applying the law of total expectation,

$$\mathbb{E}\left[\sum_{t=1}^{\tau} \Phi'(Q_\rho(t,b^*))\eta_\rho(\mathbb{E}_v(\check{c}_t(v,b^*)] - \rho)^+\right]$$

$$=\mathbb{E}\left[\sum_{t=1}^{\tau} \Phi'(Q_\rho(t,b^*))\eta_\rho(\mathbb{E}_v[c(v,b^*)] - \rho + \mathbb{E}_v[\check{c}_t(v,b^*) - c(v,b^*)])^+\right]$$

$$\leqslant\mathbb{E}\left[\sum_{t=1}^{\tau} \Phi'(Q_\rho(t,b^*))\eta_\rho[\check{c}_t(v,b^*) - c(v,b^*)]^+|\omega\right]$$

$$+ \mathbb{E}\left[\sum_{t=1}^{\tau} \Phi'(Q_\rho(t,b^*))\eta_\rho[\check{c}_t(v,b^*) - c(v,b^*)]^+|\bar{\omega}\right]$$

$$\leqslant 2T^2\eta_\rho C_{\max}^2 p,$$

where the last step uses $\Phi(x) = x^2$ so that $\Phi'(x) = 2x$, the bound of cost function, the fact that $Q_\rho(t) \leqslant Q_\rho(T) \leqslant TC_{max}$ and $\mathbb{P}(\bar{\omega}) \leqslant p$ to uniformly bound the summand on $\bar{\omega}$. Choosing $p = T^{-4}$ yields the final results.

### E.3 PROOF OF LEMMA 2

**Lemma 7 (Restatement of Lemma 2)** *Under the L2FOB algorithm, for any $t \in [T]$,*
$$\mathbb{E}[Q_\rho(t)] \leqslant C_q, \ \mathbb{E}[Q_\gamma(t)] \leqslant C_q,$$

*where*

$$C_q = \left(5TR_{max} + 2\eta_\rho C_{max}^2/T^2 + 2\eta_\gamma\gamma C_{max}^2/T^2\right)^{1/2}.$$

Applying Lemma 1, taking expectations, setting $b = b^*$, and summing the resulting inequality over $t$ yields

$$\mathbb{E}\left[\sum_{s=1}^{t}(r(v_s,b^*) - r(v_s,b_s))\right] + \mathbb{E}[\Phi(Q_\rho(t))] + \mathbb{E}[\Phi(Q_\gamma(t))]$$

$$\leqslant 4\sum_{s=1}^{t}\varepsilon_s^r + \mathbb{E}\left[\sum_{s=1}^{t}\Phi'(Q_\rho(s,b^*))\eta_\rho(\mathbb{E}_v[\check{c}_s(v,b^*)] - \rho)^+\right]$$

$$+ \mathbb{E}\left[\sum_{s=1}^{t}\Phi'(Q_\gamma(s,b^*))\eta_\gamma\mathbb{E}_v[\gamma\check{c}_s(v,b^*) - \hat{r}_s(v,b^*)]^+\right]$$

$$\leqslant 4\mathcal{E}_r(T,p) + + 2\eta_\rho C_{max}^2/T^2 + 2\eta_\gamma\gamma C_{max}^2/T^2,$$

where the last inequality follows from Lemma 3 together with the preceding arguments. Rearrange the inequality, and we obtain

$$\mathbb{E}[\Phi(Q_\rho(t))] + \mathbb{E}[\Phi(Q_\gamma(t))]$$

$$\leqslant 4\mathcal{E}_r(T,p) + 2\eta_\rho C_{max}^2/T^2 + 2\eta_\gamma\gamma C_{max}^2/T^2 + \mathbb{E}\left[\sum_{s=1}^{t}(r(v_s,b_s) - r(v_s,b^*))\right]$$

$$\leqslant 4\mathcal{E}_r(T,p) + 2\eta_\rho C_{max}^2/T^2 + 2\eta_\gamma\gamma C_{max}^2/T^2 + TR_{max},$$

We let the potential function $\Phi(x) = x^2$, then this inequality yields that

$$\mathbb{E}[Q_\rho(t)]^2 + \mathbb{E}[Q_\gamma(t)]^2 \leqslant \mathbb{E}[Q_\rho(t)^2] + \mathbb{E}[Q_\gamma(t)^2]$$

$$\leqslant 4\mathcal{E}_r(T,p) + 2\eta_\rho C_{max}^2/T^2 + 2\eta_\gamma\gamma C_{max}^2/T^2 + TR_{\max}$$

$$\leqslant 5TR_{max} + 2\eta_\rho C_{max}^2/T^2 + 2\eta_\gamma\gamma C_{max}^2/T^2$$

where the first inequality holds according to Jensen's inequality, the last inequality holds due to the property of the reward function. Since $\mathbb{E}[Q_\rho(t)]^2 \geqslant 0$ and $\mathbb{E}[Q_\gamma(t)]^2 \geqslant 0$, then by setting $C_q = (5TR_{max} + 2\eta_\rho C_{max}^2/T^2 + 2\eta_\gamma\gamma C_{max}^2/T^2)^{1/2}$, we complete the proof of Lyapunov stability.

### E.4 PROOF OF REGRET

Recall that we have the following decomposition of regret:

$$\mathcal{E}(T) = \mathbb{E}\left[\sum_{t=1}^{T} r(v_t, b^*) - \sum_{t=1}^{\tau} r(v_t, b_t)\right] \leqslant \underbrace{\nu^* \mathbb{E}[T - \tau]}_{\text{Regret after stopping}} + \underbrace{\mathbb{E}\left[\sum_{t=1}^{\tau} (r(v_t, b^*) - r(v_t, b_t))\right]}_{\text{Regret before stopping}}$$

We first derive the result for regret before stopping. Recall Lemma 1, let $b = b^*$ and rearrange the inequality, we have

$$\hat{r}_t(v_t, b^*) - \hat{r}_t(v_t, b_t)$$
$$\leqslant 4\varepsilon_t^r + \Phi'(Q_\rho(t, b^*))\eta_\rho(\mathbb{E}_v[\check{c}_t(v, b^*)] - \rho)^+ + \Phi'(Q_\gamma(t, b^*))\eta_\gamma \mathbb{E}_v[\gamma\check{c}_t(v, b^*) - \hat{r}_t(v, b^*)]^+$$
$$\Phi(Q_\rho(t-1)) - \Phi(Q_\rho(t)) + \Phi(Q_\gamma(t-1)) - \Phi(Q_\gamma(t)).$$

Summing up the above inequality over $T$ and taking the expectation, we have

$$\mathbb{E}\left[\sum_{t=1}^{\tau}(r(v_t, b^*) - r(v_t, b_t))\right]$$

$$\leqslant 4\sum_{t=1}^{\tau}\varepsilon_t^r + \mathbb{E}\left[\sum_{t=1}^{\tau}\Phi'(Q_\rho(t, b^*))\eta_\rho(\mathbb{E}_v[\check{c}_t(v, b^*)] - \rho)^+\right] + \mathbb{E}\left[\Phi(Q_\rho(0)) + \Phi(Q_\gamma(0))\right]$$

$$+ \mathbb{E}\left[\sum_{t=1}^{\tau}\Phi'(Q_\gamma(t, b^*))\eta_\gamma(\mathbb{E}_v[\gamma\check{c}_t(v, b^*) - \hat{r}_t(v, b^*)])^+\right]$$

$$\leqslant 4\sum_{t=1}^{\tau}\varepsilon_t^r + \mathbb{E}\left[\sum_{t=1}^{\tau}\Phi'(Q_\rho(t, b^*))\eta_\rho(\mathbb{E}_v[\check{c}_t(v, b^*)] - \rho)^+\right]$$

$$+ \mathbb{E}\left[\sum_{t=1}^{\tau}\Phi'(Q_\gamma(t, b^*))\eta_\gamma(\mathbb{E}_v[\gamma\check{c}_t(v, b^*) - \hat{r}_t(v, b^*)])^+\right]$$

where the first inequality follows by telescoping the potential functions, the second by the initialization $Q_{\{\rho,\gamma\}}(0) = 0$ and $\Phi(0) = 0$. Then we can apply Lemma 3 to derive that

$$\mathbb{E}\left[\sum_{t=1}^{\tau}(r(v_t, b^*) - r(v_t, b_t))\right] \leqslant 4\sum_{t=1}^{\tau}\varepsilon_t^r + \mathbb{E}\left[\sum_{t=1}^{\tau}\Phi'(Q_\rho(t, b^*))\eta_\rho(\mathbb{E}_v[\check{c}_t(v, b^*)] - \rho)^+\right]$$

$$+ \mathbb{E}\left[\sum_{t=1}^{\tau}\Phi'(Q_\gamma(t, b^*))\eta_\gamma(\mathbb{E}_v[\gamma\check{c}_t(v, b^*) - \hat{r}_t(v, b^*)])^+\right]$$

$$\leqslant 4\mathcal{E}_r(T, p) + 2\eta_\rho C_{max}^2/T^2 + 2\eta_\gamma\gamma C_{max}^2/T^2$$

$$= O(\mathcal{E}_r(T, p)).$$

For regret after stopping, Recall the definition of stopping time $\tau$ that

$$\tau = \underset{\tau' \in [T]}{\text{argmin}}\left\{\tau' \mid \sum_{t=1}^{\tau'} c(v_t, b_t) \geqslant B\right\},$$

which gives

$$\sum_{t=1}^{\tau}(\check{c}_t(v_t, b_t) - \rho) + \tau\rho + \sum_{t=1}^{\tau}(c_t(v_t, b_t) - \check{c}_t(v_t, b_t)) \geqslant B,$$

Taking the expectation and rearranging the inequality, we have

$$\mathbb{E}[T - \tau] = \frac{1}{\rho}\mathbb{E}[B - \tau\rho]$$

$$\leqslant \frac{1}{\rho} \mathbb{E}\left[\sum_{t=1}^{\tau}(\check{c}_t(v_t, b_t) - \rho) + \sum_{t=1}^{\tau}(c_t(v_t, b_t) - \check{c}_t(v_t, b_t))\right]$$

$$\leqslant \frac{1}{\rho}\left(\frac{1}{\eta_\rho}\mathbb{E}[Q_\rho(\tau)] + \mathcal{E}_c(T, p)\right)$$

$$\leqslant \frac{1}{\rho}\left(\frac{C_q}{\eta_\rho} + \mathcal{E}_c(T, p)\right),$$

where the second inequality holds since in (11), we show that

$$\eta_\rho \sum_{t=1}^{\tau}(\mathbb{E}_v[\check{c}_t(v, b_t)] - \rho)^+ \leqslant Q_\rho(\tau).$$

Applying the law of expectation indicates that

$$\mathbb{E}\left[\sum_{t=1}^{\tau}(\check{c}_t(v_t, b_t) - \rho)\right] = \mathbb{E}\left[\sum_{t=1}^{\tau}(\mathbb{E}_v[\check{c}_t(v, b_t)] - \rho)\right]$$

$$\leqslant \mathbb{E}\left[\sum_{t=1}^{\tau}(\mathbb{E}_v[\check{c}_t(v, b_t)] - \rho)^+\right]$$

$$\leqslant \frac{1}{\eta_\rho}\mathbb{E}[Q_\rho(\tau)].$$

This implies that the cumulative estimated violation is upper bounded by the terminal virtual queue, then we can apply Lemma 2 to derive the last inequality. Combining all these terms, we complete the proof of regret bound in Theorem 1.

### E.5 PROOF OF CONSTRAINT VIOLATION

Recall in (11), we have the following results for estimation violation of ROI constraint,

$$\eta_\gamma \sum_{s=1}^{t}(\mathbb{E}_v[\gamma\check{c}_t(v, b_s) - \hat{r}_s(v, b_s)])^+ \leqslant Q_\gamma(t).$$

For constraint violation, we can derive the following decomposition that

$$\mathbb{E}\left[\sum_{s=1}^{t}\gamma c(v_s, b_s) - r(v_s, b_s)\right]$$

$$= \mathbb{E}\left[\sum_{s=1}^{\tau}\mathbb{E}_v[\gamma c(v, b_s) - r(v, b_s)]\right]$$

$$\leqslant \mathbb{E}\left[\sum_{s=1}^{t}(\mathbb{E}_v[\gamma c(v, b_s) - r(v, b_s)])^+\right]$$

$$\leqslant \mathbb{E}\left[\sum_{s=1}^{\tau}(\mathbb{E}_v[\gamma\check{c}_s(v, b_s) - \hat{r}_s(v, b_s)])^+ + \sum_{s=1}^{t}(\mathbb{E}[\gamma(c(v, b_s) - \check{c}_s(v, b_s))])^+\right]$$

$$+ \mathbb{E}\left[\sum_{s=1}^{t}(\mathbb{E}_v[\hat{r}_s(v, b_s) - r(v, b_s)])^+\right]$$

$$\leqslant \mathbb{E}\left[\sum_{s=1}^{t}(\mathbb{E}_v[\gamma\check{c}_s(v, b_s) - \hat{r}_s(v, b_s)])^+\right] + \mathcal{E}_r(T, p) + \gamma\mathcal{E}_c(T, p)$$

$$\leqslant \frac{\mathbb{E}[Q_\gamma(t)]}{\eta_\gamma} + \mathcal{E}_r(T, p) + \gamma\mathcal{E}_c(T, p).$$

$$\leqslant \frac{C_q}{\eta_\gamma} + \mathcal{E}_r(T, p) + \gamma\mathcal{E}_c(T, p).$$

Then, applying Lemma 2, we can derive the constraint violation, which completes the proof of Theorem 1.

STATEMENT ON LARGE LANGUAGE MODELS (LLMS) USAGES

The large language model was used *only* as a general-purpose writing aid for English grammar checking and minor polishing. The LLM did *not* contribute to research ideation, problem formulation, algorithm design, experiment setup, analysis, or substantive writing of any technical content. All methods, results, and conclusions were conceived, written, and verified by the authors.

