# OpenReview forum: "Towards Safe and Optimal Online Bidding: A Modular Look-ahead Lyapunov Framework"
_ICLR.cc/2026/Conference — ICLR 2026 Poster_

### Official Review · Reviewer_3r7f · 2025-11-01

**Soundness:** 4
**Presentation:** 3
**Contribution:** 4
**Rating:** 8
**Confidence:** 3

**Summary:**

This paper provides a framework and an algorithm for ROI and budget constrained bidding in repeated auctions.
The framework relies on the notions of look-ahead virtual queue tracing, instead of using a more standard primal-dual approach.
The results are expressed in term of error made by prediction oracles.
The framework is used to revisite previous works.

**Strengths:**

The topic of budget and ROI constrained bidding  has already attracted a lot of attention,
and I find the framework proposed by the authors practical and intuitive (compared too, for instance, dual approaches).

**Weaknesses:**

* The writing is not always clear, and I urge the authors to check each paragraph clarity.
* What would, IMHO, what could really improve the paper would be:
1. to clarify the positioning with respect to the other papers on the topic (stochastic vs adversarial env, slack assumtions...). A table would be very welcome.
2. to better explain what the Lyapunov Framework is in general, it feels like some references and explanation could be added on the topic. For instance, could the authors tell the readers where the name comes from?

* the practicality of the assumption should be more discussed. For instance, how do you compute the "E(...)" in the algorithm in practice?

**Questions:**

* see weakness

---

> ### Author Response · Authors · 2025-11-20
> **Response to Reviewer 3r7f**
>
> We sincerely thank the reviewer for the positive assessment and the concrete suggestions. We respond to the suggestions and list the revisions we will make:
>
> * **Comparison Table.**
>     We thank the reviewer for this suggestion. We have added a comparison table in the revised manuscript to clarify the positioning of our work relative to prior studies.
>
> ---
>
> ### Table 1: Our results and related work in the first-price auction.
>
> | **Reference** | **Budget constraint** | **ROI constraint** | **Regret** | **ROI violation** | **Budget regime** | **Slater's condition free** | **Generalization** |
> | :---: | :---: | :---: | :---: | :---: | :---: | :---: | :---: |
> | [R1] | $\checkmark$ | $\boldsymbol{\times}$ | $\tilde{\mathcal O}\left((1+\tfrac{\nu^*}{\rho^2})\sqrt{T}\right)$ | $\boldsymbol{\times}$ | $\Omega(T)$ | $\times$ | $\times$ |
> | This work | $\checkmark$ | $\checkmark$ | $\tilde{\mathcal O}\left((1+\tfrac{\nu^*}{\rho})\sqrt{T}\right)$ | $\tilde{\mathcal O}(\sqrt{T})$ | $\Omega(\sqrt{T})$ | $\checkmark$ | $\checkmark$ |
>
> ---
>
> ### Table 2: Our results and related work in online bidding under uncertainty.
>
> | **Reference** | **Budget constraint** | **ROI constraint** | **Regret** | **ROI violation** | **Budget regime** | **Slater's condition free** | **Generalization** |
> | :---: | :---: | :---: | :---: | :---: | :---: | :---: | :---: |
> | [R2] | $\checkmark$ | $\checkmark$ | $\tilde{\mathcal O}\left(\sqrt{T \gamma^T_{1:N}}\right)$ | $\times$ | $\times$ | $\checkmark$ | $\times$ |
> | This work | $\checkmark$ | $\checkmark$ | $\tilde{\mathcal O}\left((1+\tfrac{\nu^*}{\rho})\sqrt{T \gamma^T_{1:N}}\right)$ | $\tilde{\mathcal O}(\sqrt{T\gamma^T_{1:N}})$ | $\Omega(\sqrt{T})$ | $\checkmark$ | $\checkmark$ |
>
> *Note: $\gamma^T_{1:N}:=\sum_{j=1}^{N} \gamma_{j, T}$.*
>
> ---
>
> ### Table 3: Our results and related work in the contextual bandit.
>
> | **Reference** | **Budget constraint** | **ROI constraint** | **Regret** | **ROI violation** | **Budget regime** | **Slater's condition free** | **Generalization** |
> | :---: | :---: | :---: | :---: | :---: | :---: | :---: | :---: |
> | [R3] | $\checkmark$ | $\times$ | $\tilde{\mathcal O}\left((1+\tfrac{\nu^*}{\rho})\sqrt{T}\right)$ | $\times$ | $\Omega(\sqrt{T})$ | $\checkmark$ | $\times$ |
> | This work | $\checkmark$ | $\checkmark$ | $\tilde{\mathcal O}\left((1+\tfrac{\nu^*}{\rho})\sqrt{T}\right)$ | $\tilde{\mathcal O}(\sqrt{T})$ | $\Omega(\sqrt{T})$ | $\checkmark$ | $\checkmark$ |
>
> ---
>
> * **Lyapunov Analysis.** Lyapunov drift analysis has been widely used to study the stability of control policies in stochastic queueing networks (see [R4]). A policy is called stable when the induced queue lengths remain finite or uniformly bounded, and smaller queues typically indicate better performance. Modern analytical frameworks providing upper bounds on queue lengths to ensure stability are developed in [R5]. For a broader introduction to Lyapunov optimization and its applications, we refer readers to [R6, R7]. We appreciate the constructive suggestion and will include these discussions to give readers a concise overview of the Lyapunov approach.
>
> * **Practicality of assumptions.**
>     In practical applications, when the context distribution is known, the expectations in Algorithm 1 can be computed either exactly via integration or approximated using Monte Carlo sampling.
>     When it is unknown, a common strategy is to estimate the distribution from data and plug these estimates into the algorithm, as in [R8, R9]. In addition, our experiments indicate that the algorithm already performs strongly when it only uses the current observed context at each round, without relying on an explicit model of the distribution.
>
> ***
>
> ### Reference
>
> [R1] "Learning to bid in repeated first-price auctions with budgets", ICML 2023.
>
> [R2] "Safe online bid optimization with return on investment and budget constraints", KDD 2025.
>
> [R3] "Triple-optimistic learning for stochastic contextual bandits with general constraints", ICML 2025.
>
> [R4] "Hitting-time and occupation-time bounds implied by drift analysis with applications.", Advances in Applied Probability, 1982.
>
> [R5] "Performance of multiclass markovian queueing networks via piecewise linear lyapunov functions." Annals of Applied Probability, 2001.
>
> [R6] "Stochastic network optimization with application to communication and queueing systems." 2022.
>
> [R7] "Communication networks: An optimization, control and stochastic networks perspective." 2014.
>
> [R8] "Contextual Bandits with Knapsacks for a Conversion Model", NeurIPS 2022.
>
> [R9] "Contextual Decision-Making with Knapsacks Beyond the Worst Case", NeurIPS 2024.

---

> > ### Comment · Reviewer_3r7f · 2025-11-26
> >
> > Thank you for your answer, I will keep my positive score.

---

> > > ### Author Response · Authors · 2025-11-26
> > >
> > > Thank you very much for your acknowledgment and constructive feedback. We sincerely appreciate the time and effort you dedicated to reviewing our work!

---

### Official Review · Reviewer_mJ77 · 2025-11-04

**Soundness:** 3
**Presentation:** 3
**Contribution:** 3
**Rating:** 4
**Confidence:** 2

**Summary:**

This paper formulates the general online bidding problem as an online learning problem with both the budget and target return-on-investment (ROI) constraints. Under this general learning problem, the authors proposed a primal–dual algorithm using a Lyapunov perspective, and they also show an regret upper bound for this algorithm with no budget overuse but some ROI violations. Additionally, for several specific settings, this upper bound matches the optimal regret order.

**Strengths:**

1. To my best knowledge, the idea of using Lyapunov function is novel.

2. The theoretical result holds in a very general framework, and also achieves strong performances in several specific but important settings.

3. Numerical results are also included in the appendix to illustrate the performance.

**Weaknesses:**

1. My main concern is about the compatibility of the learning oracle and the proposed algorithm with partial information feedback (bandit setting). Specifically, when showing sqrt{T} results in Section 5, the authors state that bounds on \Epsilon(T,p) can be obtained following some other papers. However, their arm selection process might be different from this paper. How to guarantee that the bound still holds? Could the authors elaborate more on these bounds and provide a more detailed version of the proofs? I will raise my score is this question is addressed.

2. It seems the algorithm works when only one budget constraint exists. This assumption may limit the applicability. I wonder whether a similar statement holds when there are multiple budget constraints, especially Lemma 3?

**Questions:**

See the weaknesses section.

---

> ### Author Response · Authors · 2025-11-20
> **Response to Reviewer mJ77**
>
> We appreciate the reviewer's comments and want to address your major concerns below.
>
> * **Compatibility of the learning oracle.**
>     Sorry for the confusion, and thank you for raising this point. Our method only requires reward and cost predictors equipped with **confidence intervals** that remain valid under **adaptive sampling**. Using these confidence intervals, we construct optimistic reward and pessimistic cost estimates and, together with our Lyapunov drift analysis, obtain guarantees that are adaptive to the oracle errors and therefore apply across different environments. Such oracle error bounds are standard in the online learning literature, including the works referenced in Section 5, where they are typically derived from martingale concentration inequalities such as Azuma–Hoeffding or Freedman.
>
>     Although the realized estimation error does depend on the data distribution induced by the policy, the oracle error bounds we assume are **uniform** over all action selection rules satisfying the boundedness and noise conditions. Hence, these bounds are effectively independent of the specific sampling strategy. For example, in the classical stochastic multi-armed bandit setting with a UCB algorithm, one constructs for each arm a confidence interval around its empirical mean reward using Azuma–Hoeffding or related martingale inequalities; these intervals remain valid even though the arm-selection rule is adaptive, and the resulting oracle error bounds depend only on bounded rewards and noise assumptions, not on the specific strategy. We will make these conditions explicit in the revision.
>
>     Similarly, in the results in Section 5, we will include a sketch explanation of how the learning-oracle errors are derived in each corresponding setting and why these bounds continue to hold under our algorithm.
>
> * **Multiple budget constraints.** Thanks for your insightful comment. While our problem formulation assumes a single budget constraint for simplicity, our framework naturally extends to multiple constraints. Following [R1], this can be achieved by introducing a separate virtual queue $Q_{\rho}^{(k)}$ for each budget constraint. In this case, the main guarantees become
>     $$\mathrm{Regret}(T) = O\left(\mathcal{E}\_r(T,p) + \frac{\nu^* K}{\rho}\mathcal{E}\_c(T,p)\right),$$
>     $$\mathrm{V}\_{\mathrm{ROI}}(t) = O\left(K\big(\mathcal{E}\_r(T,p) + \mathcal{E}\_c(T,p)\big)\right),\quad \forall t \in [T],$$
>     where $K$ denotes the number of constraints. Since $K$ is a fixed constant, this extension only affects constant factors and does not change our qualitative results.
>
> We hope our responses have clarified the reviewer's concerns and that the reviewer can re-evaluate our work. Please feel free to share any additional feedback, and we will address it thoroughly.
>
> ***
>
> ### Reference
>
> [R1] "On Stochastic Contextual Bandits with Knapsacks in Small Budget Regime", ICLR 2025.

---

### Official Review · Reviewer_sRuk · 2025-11-04

**Soundness:** 2
**Presentation:** 2
**Contribution:** 2
**Rating:** 4
**Confidence:** 4

**Summary:**

The paper study the online bidding problem under budget (hard) and ROI constraints, in stochastic contextual settings. The authors propose an algorithm based on a Lyapunov framework which attains regret and violations that scale linearly in the error of some online learning oracles which are assumed to exist and are used to estimate the reward and the constraints functions. Finally, the authors show how the setting they study generalise many well known problems (e.g., first price auctions) and the specific guarantees of the algorithm in the aforementioned settings.

**Strengths:**

The main strength of the paper is the setting studied. Indeed it generalises many "online bidding" problems, thus being of interest for the community. Second, I find particularly interesting the fact that the algorithm presented does not require Slater's condition to obtain the desired bound.

**Weaknesses:**

I have the following concerns on the paper:

1. First, the authors miss a fundamental related work ([1]). In [1], the authors study an online learning problem with budget and roi constraints as in this paper. The only difference is that the authors study a non-contextual setting. Moreover, [1] requires some form of Slater's condition on the ROI constraint to obtain the optimal bounds. Nonetheless, [1] attains guarantees for adversarial settings, too.
2. I find peculiar that the context distribution is assumed to be known. I believe it is not standard in the literature of contextual bandits, while it seems fundamental for the employment of the algorithm, since the expectation on the context distribution is computed at each round.
3. Paragraph "Assumptions and Baselines" is not formal enough. Specifically, Equation (2) is meaningless as it is. First, it is not clear over which distribution the expectation is taken. Moreover, if the expectation is taken over the context distribution, as it seems given that $v_t$ is not used in (2), the problem collapse to a non contextual online learning problem.
4. Throughout the paper, the authors claim that they bound the positive ROI violation (using $()^+$). Nonetheless, the standard metric is used in the theorems.
5. Finally, I don't understand the proof step at line 299. I think the inequality holds paying a $\sqrt T$ factor thanks to the Azuma inequality. Instead, the authors claims that it works as it is.

[1] "Online Learning under Budget and ROI Constraints via Weak Adaptivity", ICML 2024

**Questions:**

See weaknesses.

---

> ### Author Response · Authors · 2025-11-20
> **Response to Reviewer sRuk**
>
> We sincerely appreciate the reviewer’s constructive comments and address the main concern below.
>
> * **Related works.** Thank you for pointing out this reference. We will definitely incorporate it and provide a detailed comparison. The work [1] also studies online learning with both budget and ROI constraints and derives adaptive guarantees. We will discuss the differences in the problem setting, the assumptions, and the methodology. For example:
>     * [1] considers a non-contextual online learning problem, whereas our work operates in a **contextual** setting.
>     * [1] attains optimal rates under a Slater-type ROI margin and uses weak adaptivity to avoid **knowing** the margin a priori. In contrast, our approach does not assume a Slater margin, which broadens applicability when no margin exists.
>     * [1] builds on a refined primal–dual framework; in contrast, we employ a novel one-step look-ahead rule with a Lyapunov analysis, yielding an **anytime hard/positive** cumulative violation guarantee, whereas [1] provides only a terminal violation bound.
>
> * **Usage of the context distribution.** Though not ideal, we want to argue that the context distribution is common in several constrained contextual bandit studies. For example, [R1, R2] directly assume prior knowledge of the context distribution, whereas [R3, R4] estimate the distribution from data and then plug these estimates into their algorithms.
>
>     Moreover, we require the context distribution because we would use the rectified operator to obtain a stronger hard ROI violation guarantee. This dependence is very likely to be removed by dropping the rectified operator and working with the standard cumulative violation metric instead. We will add a detailed explanation of this trade-off in the revision.
>
> * **Assumptions and Baselines.**
>     We will carefully review all definitions in this section to ensure full formality and precision. Regarding equation (2), we would like to clarify that, although the expectation is taken with respect to the context distribution, the definition remains meaningful because the optimal solution is based on the statistical structure of the context, so that it wouldn't be reduced to a non-contextual problem, and the algorithm design also requires the context realization at every round to make a decision.
>
> * **ROI violation metric.** Thanks for pointing that out. We emphasize that, due to the rectified design, L2FOB inherently achieves a stronger hard/positive ROI violation, which upper-bounds the standard cumulative violation. However, we use the standard cumulative violation metric in the main theorem, as it is more common in the ROI literature. Thank you for catching this inconsistency. We will explicitly define and use the hard violation in the main theorem and align the notation throughout the paper to ensure clarity.
>
> * **Proof details.** Apologize for the confusion. The step at line 299 is valid because our ROI violation metric is defined in expectation (at the population level), so no Azuma Hoeffding term is required. For the realized violation, we agree that applying Azuma Hoeffding or Freedman is appropriate and introduces an additional $\tilde{O}(\sqrt{T})$ term. This does not change the guarantees, since the typical learning errors are already $\tilde{O}(\sqrt{T})$. We will clarify this in the revision.
>
> We hope that our response addresses the reviewer's concerns and that the reviewer can re-evaluate our work. Please let us know if you have any further comments, and we will try our best to address them.
>
> ***
>
> ### Reference
>
> [1] "Online Learning under Budget and ROI Constraints via Weak Adaptivity", ICML 2024.
>
> [R1] "Triple-Optimistic Learning for Stochastic Contextual Bandits with General Constraints", ICML 2025.
>
> [R2] "Resourceful Contextual Bandits", COLT 2014.
>
> [R3] "Contextual Bandits with Knapsacks for a Conversion Model", NeurIPS 2022.
>
> [R4] "Contextual Decision-Making with Knapsacks Beyond the Worst Case", NeurIPS 2024.
>
> [R5] "On Stochastic Contextual Bandits with Knapsacks in Small Budget Regime", ICLR 2025.
>
> [R6] "Stochastic Constrained Contextual Bandits via Lyapunov Optimization Based Estimation to Decision Framework", COLT 2025.
>
> [R7] "Learning to Bid in Repeated First-Price Auctions with Budgets", ICML 2023.

---

> > ### Comment · Reviewer_sRuk · 2025-11-24
> >
> > I would like to thank the Authors for their responses. The answers did not particularly change my evaluation of the work since many of my comments were confirmed. Overall, I still believe that the paper needs a major revision due to the lack of formality throughout the paper.

---

> > > ### Author Response · Authors · 2025-11-25
> > >
> > > Thank you very much for the prompt feedback. We sincerely appreciate the reviewer’s time and careful reading.
> > > In our previous response we explained why the context distribution appears in our analysis (noting that it is also assumed or estimated in several constrained contextual bandit works); clarified assumptions and baselines, emphasizing that the optimal policy does depend on the context statistics and, in particular, does not reduce to a non-contextual MAB formulation; and clarified our ROI violation notion: we use an anytime hard/positive violation defined in expectation, which can be converted into a realized high-probability guarantee via Azuma’s inequality (as the reviewer suggests).
> > > We would be extremely grateful if the reviewer could indicate which specific parts they find lacking in formality. We are happy to provide precise clarifications.
> > >
> > > Thank you again for your constructive comments, and we look forward to continuing the discussion.

---

### Official Review · Reviewer_p36P · 2025-11-06

**Soundness:** 3
**Presentation:** 3
**Contribution:** 3
**Rating:** 8
**Confidence:** 4

**Summary:**

This paper studies the problem of online bidding with a budget constraint and a return-on-investment (ROI) constraint. The authors develop a general, modular framework for this problem and propose an algorithm called L2FOB. This algorithm uses a combination of optimistic reward estimates, pessimistic cost estimates, and a one-step look ahead to optimize for the bids. The authors prove adaptive bounds on regret and ROI constraint violation in terms of errors of the reward and cost estimators.

**Strengths:**

* The authors develop a modular framework and algorithm that can be applied to many specific settings (as they do in Section 5).

* The proposed algorithm uses a nice look-ahead function to bound a potential function, which they then use to bound the Lyapunov drift and resulting regret bound. This is a neat idea. This idea seems similar to one-step model predictive control in control theory.

* The bounds on regret and ROI constraint violation are adaptive in terms of the error bounds of the reward and cost estimators. The authors achieve this regret bound without Slater's condition, although the trade-off (as they note) is that instead of strict ROI constraint satisfaction they incur some violation.

**Weaknesses:**

(Combined weaknesses and questions into one section.)

* The look-ahead idea is nice. As I mentioned in the strengths section, this seems similar to one-step model predictive control in control theory. It would be nice to discuss this area of related work in the paper. On a related note, how helpful would it be to consider a $k$-step look-ahead?

* The problem setup assumes that the context distribution is known and uses it to compute the expectations over $v$ in lines 4, 5 and 7 in Algorithm 1. Is it necessary to assume a known context distribution? How does your algorithm behave if the updates are noisy - either through point estimates, empirical estimates using the history, or an approximately known context distribution?

* Can your ideas be generalized to non-bidding problems in online optimization with constraints? How well does your algorithm scale with the number of constraints?

* The ROI constraint is not satisfied strictly, although this comes with the advantage of not assuming Slater's condition.

**Questions:**

Please see the weaknesses section.

---

> ### Author Response · Authors · 2025-11-20
> **Response to Reviewer p36P**
>
> We sincerely thank the reviewer for the positive feedback. We would like to address your questions as follows.
>
> * **Relationship with MPC.** Thanks for the insightful comments. Our look-ahead design is indeed closely related in spirit to one-step model predictive control, where a Lyapunov function is used as a surrogate for long-term performance, and the controller chooses the current action by greedily optimizing a one-step prediction that combines immediate reward/cost with the change in this function.
>     We will add a dedicated discussion to make this connection and the differences more explicit in the revision.
>
>     Regarding the $k$-step look-ahead extension, the current one-step design already performs well in the online bidding setting we study. Extending the analysis to $k$-step look-ahead is non-trivial, since it would require additional prediction over multiple steps and a careful treatment of the corresponding estimation errors, which is a very interesting future work.
>
> * **Usage of context distribution.**
>     The use of the context distribution is needed for analytical reasons. In particular, to ensure that the proof step around line 884 remains valid under the rectified operator $(\cdot)^+$, our algorithm requires taking expectations with respect to the true context distribution.
>
>     If one were to replace the true distribution by an empirical estimate, as in [R1, R2], the guarantees would likely require additional conditions on context spaces and would introduce extra context estimation error terms into both regret and violation bounds.
>     We believe this is an interesting extension for future work.
>
>     Moreover, we require the true context distribution because we want to use the rectified operator to obtain a stronger hard ROI violation guarantee. This dependence can be removed by dropping the rectified operator and working with the standard cumulative violation metric instead. We will add a detailed explanation of this trade-off in the revision.
>
> * **Beyond bidding.** We believe the answer is *yes*: the template of our look-ahead design and Lyapunov analysis can be generalized beyond bidding problems. Although our main formulation focuses on a single budget constraint for simplicity, the framework extends naturally to multiple constraints. Following [R3], this can be achieved by introducing a separate virtual queue $Q_{\rho}^{(k)}$ for each budget constraint. In this case, the main guarantees become
>     $$\mathrm{Regret}(T) = O\left(\mathcal{E}\_r(T,p) + \frac{\nu^* K}{\rho}\mathcal{E}\_c(T,p)\right),$$
>     $$\mathrm{V}\_{\mathrm{ROI}}(t) = O\left(K\big(\mathcal{E}\_r(T,p) + \mathcal{E}\_c(T,p)\big)\right),\quad \forall t \in [T],$$
>     where $K$ denotes the number of constraints. Since $K$ is a fixed constant, this extension only changes constant factors and does not affect the qualitative form of our results.
>
> * **ROI violation.**
>     Thanks for pointing that out. While we do not provide a zero-violation guarantee for the ROI constraint, this trade-off allows us to drop Slater's condition assumption and makes our framework applicable to a wider class of problems. When Slater’s condition holds, it is very likely to achieve zero constraint violation by appropriately trading off regret and violation.
>
> ***
>
> ### Reference
>
> [R1] "Contextual Bandits with Knapsacks for a Conversion Model", NeurIPS 2022.
>
> [R2] "Contextual Decision-Making with Knapsacks Beyond the Worst Case", NeurIPS 2024.
>
> [R3] "On Stochastic Contextual Bandits with Knapsacks in Small Budget Regime", ICLR 2025.

---

### Author Response · Authors · 2025-12-01
**General Response**

Dear AC and reviewers,

We sincerely thank you for your time, deep engagement, and insightful suggestions throughout the review and rebuttal processes. Unfortunately, an unexpected incident prevented us from continuing the rebuttal phase. However, we believe that the exchanges we had before this interruption have already helped to address the main concerns, and we summarize the key clarifications below:

- The common concern is the assumption on the context distribution. We have clarified that, although this assumption is not ideal, it is quite **common**. For example, [R1, R2] directly assume prior knowledge of the context distribution, whereas [R3, R4] estimate the distribution from data and then plug these estimates into their algorithms. In the same spirit, our framework is flexible enough to incorporate an estimated context distribution. By imposing additional regularity conditions on the context space, it is very likely that our algorithm can be extended to yield guarantees that explicitly depend on the context estimation error, in the spirit of [R3, R4], which we view as an interesting direction for future work.
- Reviewer p36P raises the concern that our method does not guarantee a strictly zero ROI violation. We agree with this observation, but also note that our results hold **without assuming Slater’s condition**, which inherently introduces a tradeoff. When Slater’s condition holds, it is very likely to achieve zero constraint violation with the slackness, where it can trade off regret for violation.
- Reviewer sRuk's main criticism is on the formality of our definitions of baseline and violation. For that, we explained why the context distribution appears in our analysis (noting that it is also assumed or estimated in several constrained contextual bandit works); clarified assumptions and baselines, emphasizing that the optimal policy does depend on the context statistics and, in particular, **does not reduce to** a non-contextual MAB formulation; and clarified our ROI violation notion: we use an anytime violation defined in expectation, which can be converted into a realized high-probability guarantee via Azuma’s inequality (as the reviewer suggests).
- Reviewer mJ77’s main concern is the compatibility of the learning oracle, and the reviewer indicated a willingness to increase the rating if this issue is satisfactorily addressed. We clarified that our oracle assumptions are independent of the specific sampling strategy. Such oracle error bounds are standard in the online learning literature, where they are typically derived from martingale concentration inequalities such as Azuma-Hoeffding or Freedman.
- Reviewer 3r7f provided valuable suggestions on including a comparison table, adding an introduction to Lyapunov analysis, and clarifying the practicality of our assumptions, all of which we have incorporated into the revised manuscript.

We believe our look-ahead design is novel in online learning/bidding literature, and that it brings several benefits, including removing the need for Slater’s condition, providing anytime violation guarantee, and achieving improved, and in some cases state-of-the-art, performance guarantees across different settings. We sincerely appreciate the time and effort you put into evaluating our paper!

---

**References**

[R1] "Triple-Optimistic Learning for Stochastic Contextual Bandits with General Constraints", ICML 2025.

[R2] "Resourceful Contextual Bandits", COLT 2014.

[R3] "Contextual Bandits with Knapsacks for a Conversion Model", NeurIPS 2022.

[R4] "Contextual Decision-Making with Knapsacks Beyond the Worst Case", NeurIPS 2024.

---

### Meta-Review · Area_Chair_hLk9 · 2026-01-05

**Summary:**

The reviewers generally agree that the paper studies an important and well-motivated problem: online bidding under simultaneous budget and ROI constraints. The proposed look-ahead Lyapunov framework is seen as novel and conceptually appealing. Several reviewers highlight that the approach is modular and works across different bidding and feedback settings, while achieving adaptive regret and constraint violation guarantees without needing Slater’s condition. The theoretical analysis is considered solid, and the combination of look-ahead design with Lyapunov stability arguments is viewed as a meaningful contribution beyond standard primal-dual methods. Some reviewers had concerns about assumptions, clarity, and formality, but the rebuttal addressed many of these points and added useful comparisons and explanations. Overall, the paper is considered a valuable and timely contribution that fits well with ICLR.

**Reviewer Concerns:**

Most concerns were resolved in the rebuttal. The authors clarified the use of the context distribution, explained ROI violation, the role of the learning oracle, and the connection to prior work. They also added comparison tables and more discussion of the Lyapunov framework. A few reviewers still think some parts could be more formal or clearer, but these are minor and don’t affect the main results.

**Reviewer Scores:**

Reviewers who were positive kept their high scores. Some borderline reviewers stayed cautious but did not strongly oppose acceptance after the clarifications. Overall, the discussion led to a clear majority of positive evaluations supporting acceptance.

---

### Decision · Program_Chairs · 2026-01-26

Accept (Poster)